# Federated Classification in Hyperbolic Spaces via Secure Aggregation of Convex Hulls

**Saurav Prakash** *  
*Department of Electrical and Computer Engineering*  
*University of Illinois Urbana Champaign*  
*sauravp2@illinois.edu*

**Jin Sima** *  
*Department of Electrical and Computer Engineering*  
*University of Illinois Urbana Champaign*  
*jsima@illinois.edu*

**Chao Pan** *  
*Department of Electrical and Computer Engineering*  
*University of Illinois Urbana Champaign*  
*chaopan2@illinois.edu*

**Eli Chien**  
*Department of Electrical and Computer Engineering*  
*University of Illinois Urbana Champaign*  
*ichien3@illinois.edu*

**Olgica Milenkovic**  
*Department of Electrical and Computer Engineering*  
*University of Illinois Urbana Champaign*  
*milenkov@illinois.edu*

**Reviewed on OpenReview:** *https://openreview.net/forum?id=umggDfMHha*

## Abstract

Hierarchical and tree-like data sets arise in many relevant applications, including language processing, graph data mining, phylogeny and genomics. It is known that tree-like data cannot be embedded into Euclidean spaces of finite dimension with small distortion, and that this problem can be mitigated through the use of hyperbolic spaces. When such data also has to be processed in a distributed and privatized setting, it becomes necessary to work with new federated learning methods tailored to hyperbolic spaces. As an initial step towards the development of the field of federated learning in hyperbolic spaces, we propose the first known approach to federated classification in hyperbolic spaces. Our contributions are as follows. First, we develop distributed versions of convex SVM classifiers for Poincaré discs. In this setting, the information conveyed from clients to the global classifier are convex hulls of clusters present in individual client data. Second, to avoid label switching issues, we introduce a number-theoretic approach for label recovery based on the so-called integer $B_h$ sequences. Third, we compute the complexity of the convex hulls in hyperbolic spaces to assess the extent of data leakage; at the same time, in order to limit the communication cost for the hulls, we propose a new quantization method for the Poincaré disc coupled with Reed-Solomon-like encoding. Fourth, at the server level, we introduce a new approach for aggregating convex hulls of the clients based on balanced graph partitioning. We test our method on a collection of diverse data sets, including hierarchical single-cell RNA-seq data from different patients distributed across different repositories that have stringent privacy constraints. The classification accuracy of our method is up to $\sim 11\%$ better than its Euclidean counterpart, demonstrating the importance of privacy-preserving learning in hyperbolic spaces. Our implementation for the proposed method is available at `https://github.com/sauravpr/hyperbolic_federated_classification`.

---

*Saurav, Jin, and Chao contributed equally to this work.

# 1 Introduction

Learning in hyperbolic spaces, which unlike flat Euclidean spaces are negatively curved, is a topic of significant interest in many application domains (Ungar, 2001; Vermeer, 2005; Hitchman, 2009; Nickel & Kiela, 2017; Tifrea et al., 2019; Ganea et al., 2018; Sala et al., 2018; Cho et al., 2019; Liu et al., 2019; Chami et al., 2020b; Tabaghi & Dokmanić, 2021; Chien et al., 2021; Pan et al., 2023a). In particular, data embedding and learning in hyperbolic spaces has been investigated in the context of natural language processing (Dhingra et al., 2018), graph mining (Chen et al., 2022), phylogeny (Hughes et al., 2004; Jiang et al., 2022), and genomic data studies (Raimundo et al., 2021; Pan et al., 2023a). The fundamental reason behind the use of negatively curved spaces is that hierarchical data sets and tree-like metrics can be embedded into small-dimensional hyperbolic spaces with small distortion. This is a property not shared by Euclidean spaces, for which the distortion of embedding $N$ points sampled from a tree-metric is known to be $O(\log N)$ (Bourgain, 1985; Linial et al., 1995).

In the context of genomic data learning and visualization, of special importance are methods for hyperbolic single-cell (sc) RNA-seq (scRNA-seq) data analysis (Luecken & Theis, 2019; Ding & Regev, 2021; Klimovskaia et al., 2020; Tian et al., 2023). Since individual cells give rise to progeny cells of similar properties, scRNA-seq data tends to be hierarchical; this implies that for accurate and scalable analysis, it is desirable to embed it into hyperbolic spaces (Ding & Regev, 2021; Klimovskaia et al., 2020; Tian et al., 2023). Furthermore, since scRNA-seq data reports gene activity levels of individual cells that are used in medical research, it is the subject of stringent privacy constraints – sharing data distributed across different institutions is challenging and often outright prohibited. This application domain makes a compelling case for designing novel federated learning techniques in hyperbolic spaces.

Federated learning (FL) is a class of approaches that combine distributed machine learning methods with data privacy mechanisms (McMahan et al., 2017; Bonawitz et al., 2017; Sheller et al., 2020; Kairouz et al., 2021; Pichai, 2019; Li et al., 2020b). FL models are trained across multiple decentralized edge devices (clients) that hold local data samples that are not allowed to be exchanged with other communicating entities. The task of interest is to construct a global model using single- or multi-round exchanges of secured information with a centralized server.

We take steps towards establishing the field of FL in hyperbolic spaces by developing the first known mechanism for federated classification on the Poincaré disc, a two-dimensional hyperbolic space model (Ungar, 2001; Sarkar, 2012). The goal of our approach is to securely transmit low-complexity information from the clients to the server in one round to construct a global classifier at the server side. In addition to privacy concerns, "label switching" is another major bottleneck in federated settings. Switching arises when local labels are not aligned across clients. Our FL solution combines, for the first time, learning methods with specialized number-theoretic approaches to address label switching issues. It also describes novel quantization methods and sparse coding protocols for hyperbolic spaces that enable low-complexity communication based on convex hulls of data clusters. More specifically, our main contributions are as follows.

- We describe a distributed version of centralized convex SVM classifiers for Poincaré discs (Chien et al., 2021) which relies on the use of convex hulls of data classes for selection of biases of the classifiers. In our solution, the convex hulls are transmitted in a communication-efficient manner to the server that aggregates the convex hulls and computes an estimate of the bias to be used on the aggregated (global) data.

- We perform the first known analysis of the expected complexity of random convex hulls of data sets in hyperbolic spaces. The results allow us to determine the communication complexity of the scheme and to assess its privacy features which depend on the size of the transmitted convex hulls. Although convex hull complexity has been studied in-depth for Euclidean spaces (see (Har-Peled, 2011) and references therein), no counterparts are currently known for negatively curved spaces.

- We introduce a new quantization method for the Poincaré disc that can be coupled with Reed-Solomon-like centroid encoding (Pan et al., 2023b) to exploit data sparsity in the private communication setting. This component of our FL scheme is required because the extreme points of the convex hulls have to be quantized before transmission due to privacy considerations.

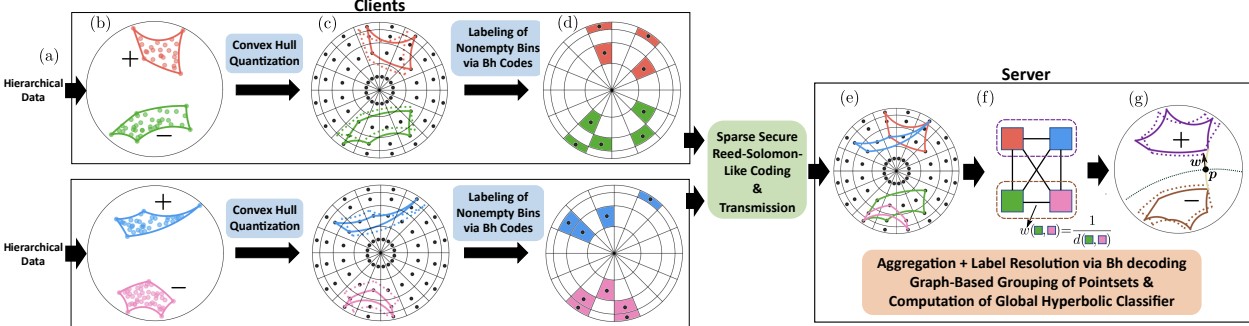

Figure 1: Diagram of the hyperbolic federated classification framework in the Poincaré model of a hyperbolic space. For simplicity, only binary classifiers for two clients are considered. (a) The clients embed their hierarchical data sets into the Poincaré disc. (b) The clients compute the convex hulls of their data to convey the extreme points to the server. (c) To efficiently communicate the extreme points, the Poincaré disc is uniformly quantized (due to distance skewing on the disc, the regions do not appear to be of the same size). (d) As part of the secure transmission module, only the information about the corresponding quantization bins containing extreme points is transmitted via Reed-Solomon coding, along with the unique labels of clusters held by the clients, selected from integer $B_h$ sequences (in this case, $h = 2$ since there are two classes). (e) The server securely resolves the label switching issue via $B_2$-decoding. (f) Upon label disambiguation, the server constructs a complete weighted graph in which the convex hulls represent the nodes while the edge weights equal $w(\cdot, \cdot) = 1/d(\cdot, \cdot)$, where $d(\cdot, \cdot)$ denotes the average pairwise hyperbolic distance between points in the two hulls. The server then performs balanced graph partitioning to aggregate the convex hulls and arrive at "proxies" for the original, global clusters. (g) Once the global clusters are reconstructed, a reference point (i.e., "bias" of the hyperbolic classifier), $\boldsymbol{p}$, is computed as the midpoint of the shortest geodesic between the convex hulls, and subsequently used for learning the "normal" vector $\boldsymbol{w}$ of the hyperbolic classifier.

- Another important contribution pertains to a number-theoretic scheme for dealing with the challenging label switching problem. Label switching occurs due to inconsistent class labeling at the clients and prevents global reconciliation of the client data sets at the server side. We address this problem through the use of $B_h$ sequences (Halberstam & Roth, 2012), with integer parameter $h \geq 2$. Each client is assigned labels selected from the sequence of $B_h$ integers which ensures that any possible collection of $h$ confusable labels can be resolved due to the unique $h$-sum property of $B_h$ integers. The proposed approach is not restricted to hyperbolic spaces and can be applied in other settings as well.

- To facilitate classification at the server level after label disambiguation, we propose a new approach for (secure) aggregation of convex hulls of the clients based on balanced graph partitioning (Kernighan & Lin, 1970).

An illustration of the various components of our model is shown in Figure 1.

The performance of the new FL classification method is tested on a collection of diverse data sets, including scRNA-seq data for which we report the classification accuracy of both the FL and global (centralized) models. The results reveal that our proposed framework offers excellent classification accuracy, consistently exceeding that of Euclidean FL learners.

It is also relevant to point out that our results constitute the first known solution for FL of hierarchical biological data sets in hyperbolic spaces and only one of a handful of solutions for FL of biomedical data (with almost all prior work focusing on imaging data (Dayan et al., 2021; Rieke et al., 2020; Mushtaq et al., 2022; Gazula et al., 2022; Saha et al., 2022)).

## 2 Related Works

**Learning in Hyperbolic Spaces.** There has been significant interest in developing theoretical and algorithmic methods for data analysis in negatively curved spaces, with a representative (but not exhaustive) sampling of results pertaining to hyperbolic neural networks (Ganea et al., 2018; Peng et al., 2021; Liu et al., 2019; Zhang et al., 2021); embedding and visualization in hyperbolic spaces (Nickel & Kiela, 2017); learning hierarchical models in hyperbolic spaces (Nickel & Kiela, 2018; Chami et al., 2020a); tree-metric denoising (Chien et al., 2022); computer vision (Khrulkov et al., 2020; Klimovskaia et al., 2020; Fang et al., 2021); Procrustes analysis (Tabaghi & Dokmanić, 2021); and most relevant to this work, single-cell expression data analysis (Klimovskaia et al., 2020; Ding & Regev, 2021), hyperbolic clustering (Xavier, 2010) and classification (Cho et al., 2019; Chen et al., 2020; Chien et al., 2021; Tabaghi et al., 2021; Pan et al., 2023a). With regards to classification in hyperbolic spaces, the work (Cho et al., 2019) initiated the study of large-margin classifiers, but resulted in a nonconvex formulation for SVMs. This issue was addressed in (Chien et al., 2021) by leveraging tangent spaces; however, the method necessitates a careful selection of the bias of classifiers, which is a nontrivial issue in the federated setting. In our work we also introduce a novel quantization scheme that is specially designed for Poincaré disc, which is distinct from the "tiling-based models" proposed in (Yu & De Sa, 2019).

**Federated Classification.** Classification has emerged as an important problem for FL. Several strategies were reported in (McMahan et al., 2017; Zhao et al., 2018; Li et al., 2020c; Karimireddy et al., 2020; Kairouz et al., 2021; Niu et al., 2023; Elkordy et al., 2022; Prakash et al., 2020b;a; Babakniya et al., 2023; Prakash et al., 2020c), with the aim to address the pressing concerns of privacy protection, non-identically distributed data, and system heterogeneity inherent in FL. In order to further alleviate the communication complexity of the training procedure, one-shot federated classification (Zhou et al., 2020; Li et al., 2020a; Salehkaleybar et al., 2021) was introduced to enable the central server to learn a global model over a network of federated devices within a single round of communication, thus streamlining the process. Despite these advances in federated classification, the intricate nature of hyperbolic spaces presents a unique challenge for the direct application of off-the-shelf FL methods. To the best of our knowledge, our work is the first one to explore specialized federated classification techniques for learning within this specific embedding domain. Notably, there are two recent papers on Riemannian manifold learning in FL setting (Li & Ma, 2022; Wu et al., 2023). While hyperbolic spaces are Riemannian manifolds, our work aims to formulate the hyperbolic SVM classification problem as a *convex optimization* problem that is solvable through Euclidean optimization methods. In contrast, Li & Ma (2022) and Wu et al. (2023) emphasize the extension of traditional Euclidean optimization algorithms, such as SVRG, to Riemannian manifolds within the FL framework. Thus, our focus is more on a new problem formulation, while Li & Ma (2022) and Wu et al. (2023) primarily address the optimization procedures. Furthermore, due to label switching, these optimization procedures are not directly applicable to our problems.

**FL for Biomedical Data Analysis.** Biomedical data is subject to stringent privacy constraints and it is imperative to mitigate user information leakage and data breaches (Cheng & Hung, 2006; Avancha et al., 2012). FL has emerged as a promising solution to address these challenges (Dayan et al., 2021; Mushtaq et al., 2022; Gazula et al., 2022; Saha et al., 2022), especially in the context of genomic (e.g., cancer genomic and transcriptomic) data analysis (Rieke et al., 2020; Chowdhury et al., 2022), and it is an essential step towards realizing a more secure and efficient approach for biological data processing. For FL techniques in computational biology, instead of the classical notions of (local) differential privacy (Dwork et al., 2014), more adequate notions of privacy constraints are needed to not compromise utility. We describe one such new notion of data privacy based on quantized minimal convex hulls in Section 4.

In what follows, we first provide a review of basic notions pertaining to hyperbolic spaces and the corresponding SVM classification task.

## 3 Classification in Hyperbolic Spaces

The Poincaré ball model (Ratcliffe et al., 1994) of hyperbolic spaces is widely used in machine learning and data mining research. In particular, the Poincaré ball model has been extensively used to address classification problems in the hyperbolic space (Ganea et al., 2018; Pan et al., 2023a).

We start by providing the relevant mathematical background for hyperbolic classification in the Poincaré ball model. Formally, the Poincaré ball $\mathbb{B}_k^n$ is a Riemannian manifold of curvature $-k$ (where $k > 0$), defined as $\mathbb{B}_k^n = \{\boldsymbol{x} \in \mathbb{R}^n : k\|\boldsymbol{x}\|^2 < 1\}$. Here and elsewhere, $\|\cdot\|$ and $\langle\cdot,\cdot\rangle$ stand for the $\ell_2$ norm and the standard inner product, respectively. For simplicity, we consider linear binary classification tasks in the two-dimensional Poincaré disc model $\mathbb{B}_k^2$, where the input data set consists of $N$ data points $\{(\boldsymbol{x}_j, y_j)\}_{j=1}^N$ with features $\boldsymbol{x}_j \in \mathbb{B}_k^2$ and labels $y_j \in \{-1, +1\}, j = 1, \ldots, N$. Our analysis can be easily extended to the multi-class classification problem and for the general Poincaré ball model in a straightforward manner.

To enable linear classification on the Poincaré disc, one needs to first define a Poincaré hyperplane, which generalizes the notion of a hyperplane in Euclidean space. It has been shown in (Ganea et al., 2018) that such a generalization can be obtained via the tangent space $\mathcal{T}_{\boldsymbol{p}}\mathbb{B}_k^2$, which is the first order approximation of the Poincaré disc at a point $\boldsymbol{p} \in \mathbb{B}_k^2$. The point $\boldsymbol{p}$ is commonly termed the "reference point" for the tangent space $\mathcal{T}_{\boldsymbol{p}}\mathbb{B}_k^2$. Any point $\boldsymbol{x} \in \mathbb{B}_k^2$ can be mapped to the tangent space $\mathcal{T}_{\boldsymbol{p}}\mathbb{B}_k^2$ as a tangent vector $\boldsymbol{v} = \log_{\boldsymbol{p}} \boldsymbol{x}$ via the logarithmic map $\log_{\boldsymbol{p}}(\cdot) : \mathbb{B}_k^2 \to \mathcal{T}_{\boldsymbol{p}}\mathbb{B}_k^2$. Conversely, the inverse mapping is given by the exponential map $\exp_{\boldsymbol{p}}(\cdot) : \mathcal{T}_{\boldsymbol{p}}\mathbb{B}_k^2 \to \mathbb{B}_k^2$. These two mappings are formally defined as follows:

$$\boldsymbol{v} = \log_{\boldsymbol{p}}(\boldsymbol{x}) = \frac{1 - k\|\boldsymbol{p}\|^2}{\sqrt{k}} \tanh^{-1}(\sqrt{k}\|(-\boldsymbol{p}) \oplus_k \boldsymbol{x}\|) \frac{(-\boldsymbol{p}) \oplus_k \boldsymbol{x}}{\|(-\boldsymbol{p}) \oplus_k \boldsymbol{x}\|}, \boldsymbol{p} \in \mathbb{B}_k^2, \boldsymbol{x} \in \mathbb{B}_k^2; \tag{1}$$

$$\boldsymbol{x} = \exp_{\boldsymbol{p}}(\boldsymbol{v}) = \boldsymbol{p} \oplus_k \left( \tanh\left( \frac{\sqrt{k}\|\boldsymbol{v}\|}{1 - k\|\boldsymbol{p}\|^2} \right) \frac{\boldsymbol{v}}{\sqrt{k}\|\boldsymbol{v}\|} \right), \boldsymbol{p} \in \mathbb{B}_k^2, \boldsymbol{v} \in \mathcal{T}_{\boldsymbol{p}}\mathbb{B}_k^2. \tag{2}$$

Here, $\oplus_k$ stands for Möbius addition in the Poincaré ball model, defined as

$$\boldsymbol{x} \oplus_k \boldsymbol{y} = \frac{(1 + 2k\langle\boldsymbol{x}, \boldsymbol{y}\rangle + k\|\boldsymbol{y}\|^2)\boldsymbol{x} + (1 - k\|\boldsymbol{x}\|^2)\boldsymbol{y}}{1 + 2k\langle\boldsymbol{x}, \boldsymbol{y}\rangle + k^2\|\boldsymbol{x}\|^2\|\boldsymbol{y}\|^2}, \forall \boldsymbol{x}, \boldsymbol{y} \in \mathbb{B}_k^2. \tag{3}$$

Furthermore, the distance between $\boldsymbol{x} \in \mathbb{B}_k^2$ and $\boldsymbol{y} \in \mathbb{B}_k^2$ is given by

$$d_k(\boldsymbol{x}, \boldsymbol{y}) = \frac{2}{\sqrt{k}} \tanh^{-1}(\sqrt{k}\|(-\boldsymbol{x}) \oplus_k \boldsymbol{y}\|). \tag{4}$$

The notions of reference point $\boldsymbol{p}$ and tangent space $\mathcal{T}_{\boldsymbol{p}}\mathbb{B}_k^2$ allow one to define a Poincaré hyperplane as

$$H_{\boldsymbol{w},\boldsymbol{p}} \triangleq \{\boldsymbol{x} \in \mathbb{B}_k^2 : \langle(-\boldsymbol{p} \oplus_k \boldsymbol{x}), \boldsymbol{w}\rangle = 0\} = \{\boldsymbol{x} \in \mathbb{B}_k^2 : \langle\log_{\boldsymbol{p}}(\boldsymbol{x}), \boldsymbol{w}\rangle = 0\}, \tag{5}$$

where $\boldsymbol{w} \in \mathcal{T}_{\boldsymbol{p}}\mathbb{B}_k^2$ denotes the normal vector of this hyperplane, resembling the normal vector for Euclidean hyperplane. Furthermore, as evident from the first of the two equivalent definitions of a Poincaré hyperplane, the reference point $\boldsymbol{p} \in H_{\boldsymbol{w},\boldsymbol{p}}$ resembles the "bias term" for Euclidean hyperplane.

**SVM in Hyperbolic Spaces**: Given a Poincaré hyperplane as defined in (5), one can formulate linear classification problems in the Poincaré disc model analogous to their counterparts in Euclidean spaces. Our focus is on hyperbolic SVMs, since they have convex formulations and broad applicability (Pan et al., 2023a). As shown in (Pan et al., 2023a), solving the SVM problem over $\mathbb{B}_k^2$ is equivalent to solving the following convex problem:

$$\min_{\boldsymbol{w} \in \mathbb{R}^2} \frac{1}{2}\|\boldsymbol{w}\|^2 \quad \text{s.t. } y_j\langle\log_{\boldsymbol{p}}(\boldsymbol{x}_j), \boldsymbol{w}\rangle \geq 1 \; \forall j \in [N]. \tag{6}$$

The hyperbolic SVM formulation in (6) is well-suited to two well-separated clusters of data points. However, in practice, clusters may not be well-separated, in which case one needs to solve the convex soft-margin SVM problem

$$\min_{\boldsymbol{w} \in \mathbb{R}^2} \frac{1}{2}\|\boldsymbol{w}\|^2 + \lambda \sum_{j=1}^N \max\left(0, 1 - y_j\langle\log_{\boldsymbol{p}}(\boldsymbol{x}_j), \boldsymbol{w}\rangle\right). \tag{7}$$

For solving the optimization problems (6) and (7), one requires the reference point $\boldsymbol{p}$. However, the reference point is not known beforehand. Hence, one needs to estimate a "good" $\boldsymbol{p}$ for a given data set before solving

the optimization problem that leads to the normal vector $\boldsymbol{w}$. To this end, for linearly separable data sets with binary labels, it can be shown that the optimal Poincaré hyperplane that can correctly classifier all data points must correspond to a reference point $\boldsymbol{p}$ that does not fall within either of the two convex hulls of data classes, since the hyperplane always passes through $\boldsymbol{p}$. Convex hulls in Poincaré disc can be defined by replacing lines with geodesics in the definition of Euclidean convex hulls (Ratcliffe et al., 2006).

To estimate the reference point $\boldsymbol{p}$ in practice, a heuristic adaptation of the Graham scan algorithm (Graham, 1972) for finding convex hulls of points in the Poincaré disc has been proposed in (Pan et al., 2023a). Specifically, the reference point is generated by first constructing the Poincaré convex hulls of the classes labeled by $+1$ and $-1$, and then choosing as the geodesic midpoint of the closest pair of extreme points (wrt the hyperbolic distance) in the two convex hulls. The normal vector $\boldsymbol{w}$ is then determined using (6) or (7).

Next, we proceed to introduce the problem of federated hyperbolic SVM classification, and describe an end-to-end solution for the problem.

## 4 Federated Classification in Hyperbolic Spaces: Problem Formulations and Solutions

As described in Section 1, our proposed approach is motivated by genomic/multiomic data analysis, since such data often exhibits a hierarchical structure and is traditionally stored in a distributed manner. Genomic data repositories are subject to stringent patient privacy constraints and due to the sheer volume of the data, they also face problems due to significant communication and computational complexity overheads.

Therefore, in the federated learning setting corresponding to such scenarios, one has to devise a distributed classification method for hyperbolic spaces that is privacy-preserving and allows for efficient client-server communication protocols. Moreover, due to certain limitations of (local) differential privacy (DP), particularly with regards to biological data, new privacy constraints are required. DP is standardly ensured through addition of privacy noise, which is problematic because biological data is already highly noisy and adding noise significantly reduces the utility of the inference pipelines. Furthermore, the proposed privacy method needs to be able to accommodate potential "label switching" problems, which arise due to inconsistent class labeling at the clients, preventing global reconciliation of the clients' data sets at the server side.

More formally, for the federated model at hand, we assume that there are $L \geq 2$ clients in possession of private hierarchical data sets. The problem of interest is to perform hyperbolic SVM classification over the union of the clients' data sets at the FL server via selection of a suitable reference point and the corresponding normal vector. For communication efficiency, we also require that the classifier be learned in a single round of communication from the clients to the server. We enforce this requirement since one-round transmission ensures a good trade-off between communication complexity and accuracy, given that the SVM classifiers only use quantized convex hulls.

In what follows, we present the main challenges in addressing the aforementioned problem of interest, alongside an overview of our proposed solutions.

1. **Data Model and Poincaré Embeddings.** We denote the data set of client $i$ which is of size $M_i$ by tuples $\boldsymbol{Z}^{(i)} = \{\boldsymbol{z}_{i,j}, y_{i,j}\}_{j=1}^{M_i}$ for $i \in \{1, \ldots, L\}$. Here, for $j \in \{1 \ldots, M_i\}$, $\boldsymbol{z}_{i,j} \in \mathbb{R}^d$ stands for the data points of client $i$, while $y_{i,j}$ denotes the corresponding labels. For simplicity, we assume the data sets to be nonintersecting (i.e., $\boldsymbol{Z}^{(i)} \cap \boldsymbol{Z}^{(j)} = \emptyset$, $\forall i \neq j$, meaning that no two clients share the same data with the same label) and generated by sampling without replacement from an underlying global data set $\boldsymbol{Z}$.

   Client $i$ first embeds $\boldsymbol{Z}^{(i)}$ into $\mathbb{B}_k^2$ (see Figure 1-(a)) to obtain $\boldsymbol{X}^{(i)} = \{\boldsymbol{x}_{i,j} \in \mathbb{B}_k^2\}_{j=1}^{M_i}$. To perform the embedding, one can use any of the available methods described in (Sarkar, 2012; Sala et al., 2018; Skopek et al., 2020; Klimovskaia et al., 2020; Khrulkov et al., 2020; Sonthalia & Gilbert, 2020; Lin et al., 2023). However, an independent procedure for embedding data points at each client can lead to geometric misalignment across clients, as the embeddings are rotationally invariant in the Poincaré disc. To resolve this problem, one can perform a Procrustes-based matching of the point sets $\boldsymbol{X}^{(i)}, i \in \{1, \ldots, L\}$ in the hyperbolic space (Tabaghi & Dokmanić, 2021). Since Poincaré embedding procedures are not the focus of this work, we make the simplifying modelling assumption that the point sets $\boldsymbol{X}^{(i)} = \{\boldsymbol{x}_{i,j} \in \mathbb{B}_k^2\}_{j=1}^{M_i}, i \in \{1, \ldots, L\}$ are sampled from a global embedding over $\cup_{i=1}^{L} \boldsymbol{Z}^{(i)}$. Although such an embedding uses the

whole data set and may hence not be private, it mitigates the need to include the Procrustes processing into the learning pipeline.

2. **Classification at the Client Level.** When performing federated SVM classification in Euclidean spaces, each client trains its own SVM classifier over its local data set; the server then "averages" the local models to obtain the global classifier (McMahan et al., 2017). It is not straightforward to extend such a solution to hyperbolic spaces. Particularly, the averaging procedure of local models is not as simple as FedAvg in Euclidean spaces due to the nonaffine operations involved in the definition of a Poincaré hyperplane (5).

We remedy the problem of aggregating classifier models at the server side by sending *minimal* convex hulls of classes instead of classifier models. A minimal convex hull is defined as follows.

**Definition 1. (Minimal Poincaré convex hull)** Given a point set $\mathcal{D}$ of $N$ points in the Poincaré disc, the minimal convex hull $CH(\mathcal{D}) \subseteq \mathcal{D}$ is the smallest nonredundant collection of data points whose convex hull contains all points in $\mathcal{D}$.

In this setting, client $i$, for $i \in \{1, \ldots, L\}$, computes the minimal convex hulls $CH_+^{(i)}$ and $CH_-^{(i)}$ of $\boldsymbol{X}_+^{(i)}$ and $\boldsymbol{X}_-^{(i)}$ respectively, as illustrated in Figure 1-(b). Here, $\boldsymbol{X}_+^{(i)}$ and $\boldsymbol{X}_-^{(i)}$ denote the point sets of the client's local classes with labels $+1$ and $-1$ respectively. To find such convex hulls, we devise a new form of the Poincaré Graham scan algorithm and prove its correctness, and in the process establish computational complexity guarantees as well. For detailed result statements, see Section 5.1. This approach is motivated by the fact that for the server to compute a suitable global reference point, it only needs to construct the convex hulls of the global clusters.

In the ideal scenario where all local labels are in agreement with the global labels and where no additional processing is performed at the client's side (such as quantization), the minimal convex hulls of the global clusters equals $CH(\cup_{i=1}^L \boldsymbol{X}_+^{(i)})$ and $CH(\cup_{i=1}^L \boldsymbol{X}_-^{(i)})$. Furthermore, the minimal convex hulls of the global and local clusters are related as $CH(\cup_{i=1}^L \boldsymbol{X}_+^{(i)}) = CH(\cup_{i=1}^L CH(\boldsymbol{X}_+^{(i)}))$ and $CH(\cup_{i=1}^L \boldsymbol{X}_-^{(i)}) = CH(\cup_{i=1}^L CH(\boldsymbol{X}_-^{(i)}))$. As a result, each client can simply communicate the extreme points of the local convex hulls to the server; the server can then find the union of the point sets belonging to the same global class, construct the global minimal convex hulls and find a global reference point $\boldsymbol{p}$.

This ideal setting does not capture the constraints encountered in practice since a) the extreme points of the convex hulls have to be quantized to ensure efficient transmission; b) the issue of label switching across clients has to be resolved; and c) secure aggregation and privacy protocols have to be put in place before the transmission of local client information.

**Poincaré Quantization.** To reduce the communication complexity, extreme points of local convex hulls are quantized before transmission to the server. In Euclidean spaces, one can perform grid based quantization, which ensures that the distance between pairs of points from arbitrary quantization bins are uniformly bounded. To meet a similar constraint, we describe next the quantization criterion for $\mathbb{B}_k^2$.

**Definition 2. ($\epsilon$-Poincaré quantization)** For $\epsilon > 0$, quantization scheme for $\mathbb{B}_k^2$ is said to be a $\epsilon$-Poincaré quantization scheme if for any pair of points $\boldsymbol{x}$ and $\boldsymbol{y}$ that lie in the same quantization bin, their hyperbolic distance $d_k(\boldsymbol{x}, \boldsymbol{y})$ as defined in (4) satisfies $d_k(\boldsymbol{x}, \boldsymbol{y}) \leq \epsilon$.

To facilitate the transmission of extreme points, we develop a novel $\epsilon$-Poincaré quantization scheme, which we subsequently refer to as $PQ_\epsilon(\cdot)$ (for a mathematical analysis of the quantization scheme, see Section 5.2). The $PQ_\epsilon(\cdot)$ method uses radial and angular information (akin to polar coordinates in Euclidean spaces). To bound the bins in the angular domain, we partition the circumference such that the hyperbolic length of any quantization bin along the circumference is bounded from above by $\epsilon/2$. Similarly, we choose to partition the radial length so that all bins have hyperbolic radial length equal to $\epsilon/2$. This ensures that the quantization criteria in Definition 2 is satisfied. The parameter $\epsilon$ determines the total number of the quantization bins, and thus can be chosen to trade the quantization error with the number of bits needed for transmitting the convex hulls. Furthermore, we choose the "bin centroids (centers)" accordingly. Using the $PQ_\epsilon(\cdot)$ protocol, each client $i$ quantizes its minimal Poincaré convex hulls $CH_+^{(i)}$ and $CH_-^{(i)}$ to create two quantized convex hull point sets $Q_+^{(i)} = \{PQ_\epsilon(\boldsymbol{x}) : \boldsymbol{x} \in CH_+^{(i)}\}$

and $Q_-^{(i)} = \{PQ_\epsilon(\boldsymbol{x}) : \boldsymbol{x} \in CH_i^{(i)}\}$. The quantized convex hulls (i.e., extreme points) are also used for measuring the information leakage of the hyperbolic classification approach, as outlined in what follows.

**Privacy Leakage.** A major requirement in FL is to protect the privacy of individual client data (Bonawitz et al., 2017). One of the most frequently used privacy guarantees are that the server cannot learn the local data statistics and/or identity of a client based on the received information (for other related privacy criteria in federated clustering, please refer to (Pan et al., 2023b)). However, such a privacy criteria is quite restrictive for classification tasks using $Q_+^{(i)}$ and $Q_-^{(i)}$, for $i \in [L]$, as the server needs to reconstruct the hulls for subsequent processing. Hence, we consider a different notion of privacy, where some restricted information pertaining to local data statistics (specifically, $Q_+^{(i)}$ and $Q_-^{(i)}$, for $i \in [L]$), is allowed to be revealed to the server, while retaining full client anonymity.

More precisely, to quantify the information leakage introduced by our scheme, we make use of the so-called $\epsilon$-minimal Poincaré convex hull, defined next.

**Definition 3. ($\epsilon$-Minimal Poincaré convex hull)** Given $\epsilon > 0$, a quantization scheme $PQ_\epsilon(\cdot)$ for $\mathbb{B}_k^2$, and a set $\mathcal{D}$ of $N$ points in the Poincaré disc, let $Q = \{PQ_\epsilon(\boldsymbol{x}) : \boldsymbol{x} \in CH(\mathcal{D})\}$ be the point set obtained by quantizing the extreme points of the convex hull of $\mathcal{D}$. The $\epsilon$-minimal Poincaré convex hull $\hat{CH}(\mathcal{D}) \subseteq Q$ is the smallest nonredundant collection of points whose convex hull contains all points in $Q$.

To avoid notational clutter, we henceforth use "quantized convex hull" of a given point set $\mathcal{D}$ to refer to the $\epsilon$-minimal Poincaré convex hull of $\mathcal{D}$. Client $i$ computes two quantized convex hulls $\hat{CH}(\boldsymbol{X}_+^{(i)})$ and $\hat{CH}(\boldsymbol{X}_-^{(i)})$ by finding the minimal convex hulls of $Q_+^{(i)}$ and $Q_-^{(i)}$, respectively. We use the number of extreme points of the quantized convex hull (i.e., the complexity of the convex hull), defined next, as a surrogate function that measures privacy leakage.

**Definition 4. ($\epsilon$-Convex hull complexity)** Given $\epsilon > 0$, a $\epsilon$-Poincaré quantization scheme $PQ_\epsilon(\cdot)$, and a set $\mathcal{D}$ of $N$ points in the Poincaré disc, let $Q = \{PQ_\epsilon(\boldsymbol{x}) : \boldsymbol{x} \in CH(\mathcal{D})\}$ be the point set obtained by quantizing the extreme points in the convex hull of $\mathcal{D}$. The $\epsilon$-convex hull complexity is the number of extreme points in the convex hull of $Q$, i.e., the size of $\hat{CH}(\mathcal{D})$.

We would like to point out that our notion of privacy does not rely on (local) differential privacy (DP) or information theoretic privacy. Instead, it may be viewed as a new form of "combinatorial privacy," enhanced by controlled data quantization-type obfuscation. This approach to privacy is suitable for computational biology applications, where DP noise may be undesirable as it significantly reduces the utility of learning methods.

In addition to compromising accuracy, DP has practical drawbacks for other practical applications (Kifer & Machanavajjhala, 2011; Bagdasaryan et al., 2019; Zhang et al., 2022; Kan, 2023). The work Bagdasaryan et al. (2019) established that DP models are more detrimental for underrepresented classes. More precisely, fairness disparity is significantly pronounced in DP models, as the accuracy drop for minority (outlier) classes is more significant than for other classes. Additionally, in many practical scenarios, meeting requirements for DP, which rely on particular assumptions about the data probability distribution, might prove challenging (Kifer & Machanavajjhala, 2011; Zhang et al., 2022; Kan, 2023). For instance, when database records exhibit strong correlations, ensuring robust privacy protection becomes difficult. All of the above properties that cause issues are inherent to biological (genomic) data. First, genomic data is imbalanced across classes. Second, as it derives from human patients, it is highly correlated due to controlled and synchronized activities of genes within cells of members of the same species.

To overcome the aforementioned concerns, we require that 1) the clients reveal a small as possible number of data points that are quantized (both for the purpose of reducing communication complexity and further obfuscating information about the data); 2) the server collects data without knowing its provenance or the identity of the individual clients that contribute the data through the use of secure aggregation of the quantized minimal convex hulls of the client information. To address the first requirement, we introduce a new notion of privacy leakage in Definition 4. If the number of quantized extreme points is small and the quantization regions are large, the $\epsilon$-convex hull complexity is small and consequentially the privacy leakage is "small" (i.e, the number of points whose information is leaked is small). For bounds on the

$\epsilon$-convex hull complexity in hyperbolic spaces (and, consequently, the privacy leakage), the reader is referred to Section 5.3.

To address the second requirement, we adapt standard secure aggregation protocols to prevent the server from inferring which convex hull points belong to which client, thus hiding the origin of the data.

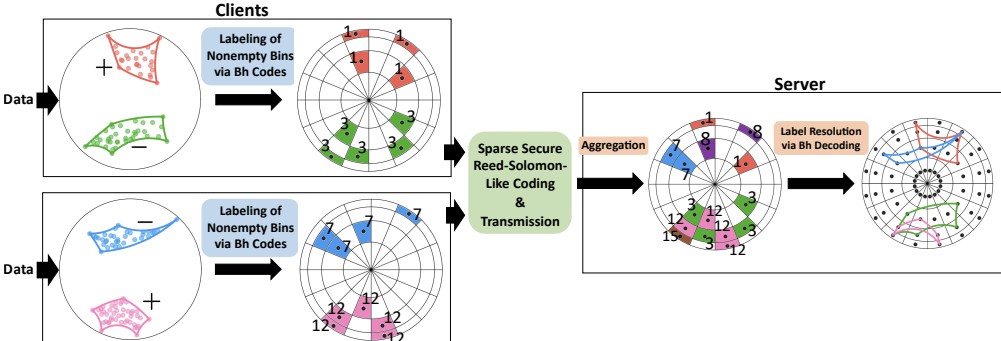

Figure 2: Proposed solution for the label switching problem. Each client has its own data set with its own set of binary labels. Upon obtaining the convex hull, the extreme points are quantized to their nearest bin-centers, and the corresponding bins are labeled with one of two integers from the $B_2$ sequence assigned to each client. At the server side, the values in the nonempty bins are first aggregated. Next, by the property of the $B_h$ sequence that the sum of any subset of size $\leq h$ of elements in the sequence is unique, all individual labels for the bins that are shared across multiple convex hulls (such as the bins colored in purple and brown) are decoded. Therefore, the server is able to recover individual (quantized) local convex hulls, which are further processed for global labeling and reference point computation (see Figure 1).

**Label Switching.** Another major challenge faced by the clients is "label switching." When performing classification in a distributed manner, unlike for the centralized case, one has to resolve the problem of local client labels being potentially switched compared to each other and/or the ground truth. Clearly, the mapping from the local labels at the client level to the global ground truth labels is unknown both to the client and the server.

The label switching problem is illustrated in Figure 2, where clients use mismatched choices for assigning the binary labels $\{-1, +1\}$ to their local data points. To enable federated classification, the label switching problem has to be addressed in a private and communication efficient manner, which is challenging. Our proposed solution is shown in Figure 2, and it is based on so-called $B_h$ sequences. In a nutshell, $B_h$ sequences are sequences of positive integers with the defining property that any sum of $\leq h$ possibly repeated terms of the sequence uniquely identifies the summands (see Section 5.4 for a rigorous exposition). Each client is assigned two different integers from the $B_h$ sequence to be used as labels of their classes so that neither the server nor the other clients have access to this information. Since secure aggregation is typically performed on sums of labels, and the sum uniquely identifies the constituent labels that are associated with the client classes, all labels can be matched up with the two ground truth classes. As an illustrative example, the integers $1, 3, 7, 12, 20, 30, 44, \dots$ form a $B_2$ sequence, since no two (not necessarily distinct) integers in the sequence produce the same sum. Now, if one client is assigned labels 1 (red) and 3 (green), while another is assigned 7 (blue) and 12 (pink), then a securely aggregated label $1 + 7 = 8$ (purple) for points in the intersection of their convex hulls uniquely indicates that these points belonged to both the cluster labeled 1 at one client, and the cluster labeled 7 at another client (see Figure 2, right panel). Note that there are many different choices for $B_h$ sequences to be used, which prevents potential adversaries to infer information about label values and assignments.

Note that a different notion of label misalignment termed "concept shift" has been considered in prior works, including (Sattler et al., 2020; Bao et al., 2023). In the referenced settings, label misalignment is a result of client-specific global conditional data distributions (i.e., distributions of global labels given local feature). Label misalignments are therefore handled by grouping clients based on the similarity of their conditional data distributions; furthermore, a different global model is trained for each group of clients. In contrast, we actually align labels of clients in the presence of switched labels, since for our problem,

the conditional global ground truth distributions are the same. Another related line of works is that of "label noise" Petety et al. (2020); Song et al. (2022), where effectively, the global label for each individual data point is assumed to be noisy, thus differing from our setting with misalignments across clients.

3. **Secure Transmission.** As shown in Figure 1-(d), the number of nonempty quantization bins labeled by $B_h$ integers is typically significantly smaller than the total number of quantization bins. As the client only needs to communicate to the server the number of extreme points in nonempty quantization bins, sparse coding protocols and secure aggregation protocols are desirable. In particular, the server should be able to collect data without knowing its provenance or the individual clients/users that contributed the data through the use of secure aggregation of the quantized minimal convex hulls of the client data, while ensuring that the secure transmission method is also communication efficient. We address this problem by leveraging a method proposed in (Pan et al., 2023b) termed Secure Compressed Multiset Aggregation (SCMA). The key idea is to use Reed-Solomon-like codes to encode information about the identity of nonempty quantization bins and the number of extreme points in them through evaluations of low-degree polynomials over prime finite field. The bin indices represent distinct elements of the finite field (since all bins are different), while the number of extreme points in a bin represents the coefficient of the term corresponding to the element of the finite field encoding the bin (the number of points in different bins may be the same).

In our FL method, we follow a similar idea, with the difference that we use the local labels (encoded using $B_h$ sequences) associated with the clusters to represent the coefficients of the terms, instead of using the number of extreme points. For this, we first index the quantization bins as $1, 2, 3, \ldots$, starting from the innermost angular bins and moving counter-clockwise (the first bin being at angle 0). For the quantization bins in Figure 2, the bins are indexed $1, \ldots, 64$. For example, the four nonempty bins corresponding to label 1 are indexed by $18, 21, 51, 53$, while the five non-empty green bins are indexed by $27, 31, 44, 46, 59$. The SCMA secure transmission procedure is depicted in Figure 3, where the clients send the weighted sum of bin indices to the server so that the weights are labels of the clusters and the server aggregates the weighted sums. From the weighted sums, the server decodes the labels of all nonempty bins. The underlying field size is chosen to be greater than the number of bins. A mathematically rigorous exposition of the results is provided in Section 5.5.

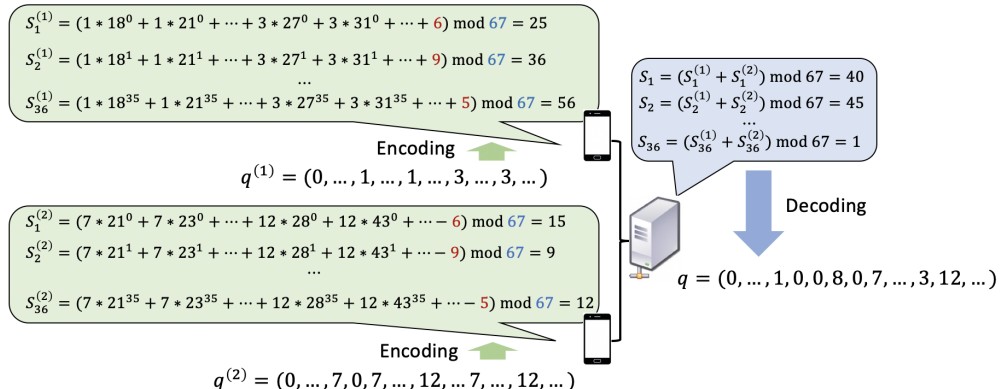

Figure 3: The encoding and decoding procedure used in SCMA for transmitting the data sets shown in Figure 2.

4. **Classification at the Server.** Upon receiving the information about quantized extreme points of the data classes communicated by the clients, the server performs the following steps to construct a global SVM classifier.

**Label Decoding.** The server uses the $B_h$ labels of nonempty quantization bins, which equal the aggregate (sum) of the corresponding local labels used by clients that contain data points in that quantization bin (see Figure 2 where shared nonempty bins are indexed by aggregated labels $8(= 1 + 7)$ and $15(= 3 + 12)$ (purple and brown respectively)). The sums of the labels for each bin are acquired during the secure transmission step. By the defining property of $B_h$ sequences, the server correctly identifies the individual

local labels that contribute to a nonempty bin from their sum, as long as the parameter $h$ is larger than or equal to the number of different local labels that contribute to the bin. Therefore, $h$ should be selected as the maximum number of classes across all clients that have at least one extreme point in the same bin. Therefore, $h \leq 2L$ in the worst case since there are at most $2L$ local classes in total. In practice, $h$ is significantly smaller (see an example for $h = 2$ in Figure 2). Thus, through $B_h$ decoding, each extreme point in each quantized convex hull can be recovered; this implies that each individual convex hull can also be unambiguously recovered at the server side, as shown in Figure 1-(e).

**Remark 4.1.** *The above steps allow the server to recover the quantized convex hulls of the local classes at each client, without learning anything about the data provenance. More precisely, the server cannot associate the identities of the clients to the recovered convex hulls. Furthermore, the server receives only a very small number of quantized extreme points of the convex hull of the data sets at the client side, and the quantization noise may be roughly viewed as a noise-privacy mechanism.*

**Balanced Grouping of Convex Hulls.** After label decoding, the next server task involves clustering the $2L$ convex hulls into 2 groups, where each group corresponds to a global ground truth label. To solve this problem, we propose a graph-theoretic approach using information about distances between pairs of point sets in the Poincaré disc. More precisely, let the convex hulls be labeled by $\{1, \ldots, 2L\}$, and let $CH^{(i)}$ denote the $i$th convex hull for $i \in \{1, \ldots, 2L\}$. We construct a weighted complete graph $\mathcal{G}(V)$ with $V = \{1, \ldots, 2L\}$ representing the $2L$ convex hulls. For a pair of nodes $u, v \in V$, the edge-weight $w(\{u, v\}) = 1/d_{\mathcal{G}}(u, v)$, where $d_{\mathcal{G}}(u, v)$ is the average pairwise hyperbolic distance between points in the convex hulls $CH^{(u)}$ and $CH^{(v)}$, i.e.,

$$d_{\mathcal{G}}(u, v) = \frac{\sum_{\boldsymbol{x}_1 \in CH^{(u)}} \sum_{\boldsymbol{x}_2 \in CH^{(v)}} d_k(\boldsymbol{x}_1, \boldsymbol{x}_2)}{|CH^{(u)}||CH^{(v)}|}. \tag{8}$$

Using the Kernighan-Lin balanced graph partitioning algorithm (Kernighan & Lin, 1970), we arrive at two groups of aggregated local convex hulls. Particularly, $V$ is partitioned into two parts, each of size $L$, so as to minimize the sum of the edge-weights between the two parts (i.e., the weight of the balanced cut). This in turn results in two global groups of convex hulls. These steps are shown in Figure 1-(f). The formal description of our approach is provided in Section 5.6.

**Learning the Global SVM Classifier.** Upon completion of the graph partitioning step, the server assigns global binary labels to the two global point sets. Then, following the procedure described in Section 3, the server constructs the global reference point $\boldsymbol{p}$. Due to quantization, the server treats the reconstructed global point sets as approximations/surrogates for the true global ground truth clusters, and solves the optimization problem (6) or (7) to obtain the normal vector $\boldsymbol{w}$ of the classifier. This step is shown in Figure 1-(g). Due to the distortions in the local hulls caused by Poincaré quantization, the estimated global hulls are also distorted, as shown in Figure 1-(g). Hence, the performance of the trained classifier depends on the quantization size $\epsilon$: A smaller value of $\epsilon$ leads to smaller convex hull distortions but larger communication overhead. In Section 6, we demonstrate through extensive simulations that the the global classifier learned using our framework performs well in practice for a relatively large range of $\epsilon$ values, both for synthetic and real data sets.

## 5   Detailed Explanations and Main Results

In what follows, we provide rigorous algorithmic results and performance guarantees for every learning and computational step of the proposed one-shot federated SVM classification algorithm for hyperbolic spaces.

### 5.1   Poincaré Convex Hulls

The first step of the end-to-end solution is to construct $CH_+^{(i)}$ and $CH_-^{(i)}$, the sets of extreme points in the convex hulls of local clusters at the clients $i \in \{1, \ldots, L\}$, with local labels $+1$ and $-1$, respectively. For this purpose, we adapt the Graham scan algorithm (Graham, 1972) widely used for computing convex hulls in Euclidean space. Our Graham scan is described in Alg. 1, while the performance guarantees and complexity of the method are summarized in the next theorem.

---

**Algorithm 1** Poincaré Graham Scan

---
1: **Input:** Points $\mathcal{D} = \{\boldsymbol{x}_1, \cdots, \boldsymbol{x}_N\}$ in the Poincaré disc $\mathbb{B}_k^2$.
2: $\boldsymbol{b} \leftarrow$ the point in $\mathcal{D}$ at largest distance from the origin of $\mathbb{B}_k^2$, i.e., $\boldsymbol{o} = [0, 0]$.
3: Normal vector $\boldsymbol{n} \leftarrow -\frac{\log_{\boldsymbol{b}}(\boldsymbol{o})}{\|\log_{\boldsymbol{b}}(\boldsymbol{o})\|}$, $\log_{\boldsymbol{b}}(\boldsymbol{o})$ is logarithmic map in (1).
4: Tangent vector $\boldsymbol{t} \leftarrow \boldsymbol{R}\,\boldsymbol{n}$, where $\boldsymbol{R} = \begin{bmatrix} 0 & -1 \\ 1 & 0 \end{bmatrix}$ denotes the $\frac{\pi}{2}$ counter-clockwise rotation matrix.
5: Sorted data $\mathcal{V} \leftarrow$ for $j \in [N]$, arrange points in $\mathcal{D}$ based on the principle angle of $\log_{\boldsymbol{b}}(\boldsymbol{x}_j)$ with respect to $\boldsymbol{t}$, in ascending order. For points with the same principle angle, keep only the one with the largest norm $\|\log_{\boldsymbol{b}}(\boldsymbol{x}_j)\|$.
6: $\mathcal{V} \leftarrow \mathcal{V} - \boldsymbol{b}$.
7: $m \leftarrow |\mathcal{V}|$.
8: STACK $= [\boldsymbol{b}]$.
9: **for** $j \in [m]$ **do**
10:    **while** len(STACK)>1 and CCW(STACK[-2], STACK[-1], $\mathcal{V}[j]$)$\leq$0 **do**
11:      STACK.pop()
12:    **end while**
13:    STACK.append($\mathcal{V}[j]$)
14: **end for**
15: **Return** STACK

---

**Algorithm 2** CCW

---
1: **Input:** Points $\boldsymbol{x}$, $\boldsymbol{t}$, $\boldsymbol{z}$ in $\mathbb{B}_k^2$.
2: $\boldsymbol{u}_1 \leftarrow \frac{\log_{\boldsymbol{x}}(\boldsymbol{t})}{\|\log_{\boldsymbol{x}}(\boldsymbol{t})\|}$.
3: $\boldsymbol{u}_2 \leftarrow \frac{\log_{\boldsymbol{x}}(\boldsymbol{z})}{\|\log_{\boldsymbol{x}}(\boldsymbol{z})\|}$.
4: **Return** $\boldsymbol{u}_1[0]\boldsymbol{u}_2[1] - \boldsymbol{u}_1[1]\boldsymbol{u}_2[0]$.

---

**Theorem 5.1.** *Given a set $\mathcal{D}$ of $N$ points in the Poincaré disc, Alg. 1 returns $CH(\mathcal{D})$ in $O(N \log N)$ time, where $CH(\mathcal{D})$ denotes the set of extreme points in the minimal convex hull of $D$.*

Based on this result, it is easy to see that Alg. 1 has the same complexity as the Graham Scan algorithm for Euclidean spaces. The proof of Theorem 5.1 can be found in Appendix A.

## 5.2 Poincaré Quantization

Quantization is standardly used to trade-off communication overhead and accuracy of point representations. In our FL setting, quantization also enables the new label encoding and decoding procedures that resolve label switching, as well as secure aggregation during transmission.

The quantization approach for Poincaré discs exploits radial symmetry of the space. Furthermore, for a given quantization parameter $\epsilon > 0$, it ensures that the hyperbolic distance between any two points in a quantization bin is bounded by $\epsilon$.

We assume that the data points are confined to a region with a Euclidean radius of $R$ on the Poincaré disc, and correspondingly within a hyperbolic radius of $R_H$. Given the curvature constant $k$, we let $s = 1/\sqrt{k}$. We can then relate $R$ and $R_H$ as follows (Hitchman, 2009):

$$R_H = s \ln\left(\frac{s+R}{s-R}\right) \quad \text{and} \quad R = s\left(\frac{e^{\frac{R_H}{s}} - 1}{e^{\frac{R_H}{s}} + 1}\right). \tag{9}$$

Furthermore, since the Poincaré disc is conformal, it preserves angles. Hence, we directly perform quantization in the hyperbolic plane, and then map the points back to the Poincaré disc. Towards this end, we start by observing that the circumference of a circle centered at origin and with hyperbolic radius $r_H$ equals

(Hitchman, 2009):

$$Cir(r_H) = 2\pi s \sinh\left(\frac{r_H}{s}\right) \tag{10}$$

We first partition the Poincaré disc in terms of concentric circles around the origin, and then for each such partition, we perform further radial partition.

Let $N_\Theta$ denote the number of angular quantization bins. Consider two points at the outer boundary of the outermost quantization bin. Then, by (10), the length along the outer boundary equals $Cir(R_H)/N_\Theta$. Bounding this length by $\epsilon/2$, we have $N_\Theta = 2Cir(R_H)/\epsilon$. Clearly, the inner boundaries of the radial bins will have a length $\leq \epsilon/2$.

Next, let $N_{R_H}$ denote the number of radial bins for any given angle. We create radial bins in such a way that their hyperbolic lengths are upper-bounded by $\epsilon/2$. Hence, the total number of quantization bins equals $N_{R_H} = 2R_H/\epsilon$. We write $\phi_{n_1} = \frac{n_1}{N_\Theta} 2\pi$, where $n_1 \in [N_\Theta]$, and for $n_2 \in [N_{R_H}]$ set

$$h_{n_2} = \frac{n_2}{N_{R_H}} R_H. \tag{11}$$

Given this notation, a quantization bin $B(n_1, n_2)$ is completely characterized by $\phi_{n_1-1}$ and $\phi_{n_1}$ in the angular domain, and $h_{n_1-1}$ and $h_{n_1}$ in the radial domain, where $\phi_0 = 0$ and $h_0 = 0$. Furthermore, any point in the bin $B(n_1, n_2)$ is mapped to the bin-center $bc(n_1, n_2)$, defined as the point that partitions the bin into four parts of the same hyperbolic radial length and angularly symmetric. As a result, the angular $\phi_{bc(n_1,n_2)}$ and radial $h_{bc(n_1,n_2)}$ coordinates of the bin-center equal

$$\phi_{bc(n_1,n_2)} = \frac{\phi_{n_1-1} + \phi_{n_1}}{2} \quad \text{and} \quad h_{bc(n_1,n_2)} = \frac{h_{n_2-1} + h_{n_2}}{2}. \tag{12}$$

To map the results from the hyperbolic plane to the Poincaré disc, the Euclidean radius of the corresponding bin-center is obtained using (9) and its 2D projection is obtained using $\phi_{bc(n_1,n_2)}$.

As an example, consider a point $\boldsymbol{x} \in \mathbb{B}_k^2$ within a bin $B(n_1, n_2)$. It is quantized to

$$\hat{\boldsymbol{x}} = (\alpha \cos(\zeta), \alpha \sin(\zeta)), \tag{13}$$

where $\zeta = \phi_{bc(n_1,n_2)}$, $\alpha = s(e^{\frac{\tau}{s}} - 1)/(e^{\frac{\tau}{s}} + 1)$ and $\tau = h_{bc(n_1,n_2)}$.

Using Alg. 1, a client $i \in \{1, \ldots, L\}$ quantizes the extreme points in the minimal convex hulls $CH_+^{(i)}$ and $CH_-^{(i)}$ to obtain the quantized point sets $Q_+^{(i)}$ and $Q_-^{(i)}$. Finally, to obtain the quantized convex hulls $\hat{CH}_+^{(i)}$ and $\hat{CH}_-^{(i)}$ (see Definition 3), the client applies Alg. 1 on $Q_+^{(i)}$ and $Q_-^{(i)}$. This ensures that the quantized convex hulls remain convex.

### 5.3 Convex Hull Complexity

Since the convex hull complexity depends on the distribution of the data and characterizing convex hulls in hyperbolic space is complicated, we do not have a general complexity analysis for arbitrary (quantized) data distribution. Instead, we analyze the convex hull complexity assuming that the data is uniformly distributed on the Poincaré disc. It can be shown that the expected convex hull complexity, i.e., the expected number of extreme points in the convex hull of any collection of $N$ points sampled independently and uniformly at random from the Poincaré disc equals $O(N^{\frac{1}{3}})$ (see Appendix D). In comparison, for a two-dimensional Euclidean space, the expected convex hull complexity also equals $O(N^{\frac{1}{3}})$, provided that the points are sampled independently and uniformly at random Har-Peled (2011). Furthermore, note that our scheme uses quantized extreme points of the convex hull, so that we need to find an upper bound on the $\epsilon$-convex hull complexity (see Definition 4), assuming that the data is uniformly distributed over the Poincaré disc. Yet, rather than directly analyzing the $\epsilon$-convex hull complexity, which is a result of constructing convex hulls, quantizing, constructing convex hulls, we have the following result that characterizes the average number of points in the minimal convex hull of $N$ quantized points sampled independently and uniformly at random from the Poincaré disc. This result serves as an upper bound on the $\epsilon$-convex hull complexity. The proof of the theorem is available in Appendix E. Recall that this complexity value also captures the privacy leakage of each client.

**Theorem 5.2.** *Let $\boldsymbol{x}_1, \ldots, \boldsymbol{x}_N$ be $N$ points uniformly distributed over a Poincaré disc of radius $R < s$, where $N$ is sufficiently large. Let $\hat{\boldsymbol{x}}_1, \ldots, \hat{\boldsymbol{x}}_N$ be quantized values of $\boldsymbol{x}_1, \ldots, \boldsymbol{x}_N$, obtained using the quantization rule described in (13), with parameter $\epsilon = O(N^{-c})$, for some constant $c > 0$. Then, the expected number of extreme points of the minimal convex hull of $\hat{\boldsymbol{x}}_1, \ldots, \hat{\boldsymbol{x}}_N$ is at most $O(N^c)$ when $c < \frac{1}{2}$, at most $O(N^{1-c})$ when $\frac{1}{2} \le c \le \frac{2}{3}$, and at most $O(N^{\frac{1}{3}})$ when $\frac{2}{3} < c$.*

### 5.4 Label Encoding

Due to the problem of label switching, different classes at different clients may have the same label and be confused at the server. To avoid this issue, we propose encoding the class labels using $B_h$ sequences, defined as follows.

**Definition 5.** *$B_h$ sequences (or Sidon sets of order $h$) are sets (sequences) of nonnegative integers such that the sums of any $h$ (with repetitions allowed) integers from the set are all different. Formally, a sequence $A = \{a_1, \ldots, a_m\}$, $0 \le a_1 < a_2 < \ldots < a_m$, is a $B_h$ sequence if for any $i_1, \ldots, i_h \in \{1, \ldots, m\}$, the multiset $\{i_1, \ldots, i_h\}$ can be uniquely determined based on the sum $a_{i_1} + \ldots + a_{i_h}$.*

$B_h$ sequences for $h = 2$ (Sidon sequences) have found numerous applications in error-correction coding (Milenkovic et al., 2006; Kovačević & Tan, 2018). For $B_h$ sequences, the question of interests is how small the $m$th term, $a_m$, can be, or conversely, how large $m$ can be given a bound on $a_m$. It can be shown that $a_m$ is at least $\frac{m^h}{h}$ since there are $m^h$ possible sums of $h$ integers in $A$ and the maximum sum $a_{m-h+1} + a_{m-h+2} + \ldots + a_m$ is at least $m^h$. Therefore, we have $ha_m \ge m^h$. This simple lower bound on $a_m$ shows that $a_m$ scales as $O(m^h)$ for fixed $h$. On the other hand, the work in (Bose & Chowla, 1960) provides constructions for $B_h$ sets $A$ where $a_m$ is at most $m^h$, and this is asymptotically optimal based on the lower bound on $a_m$.

Note that for any $B_h$ sequence $A = \{a_1, \ldots, a_m\}$, the set $A' = \{a_2 - a_1, \ldots, a_m - a_1\}$ is also a $B_h$ set (Kovacevic & Tan, 2017). Moreover, the sums $a'_{i_1} + a'_{i_2} + \ldots + a'_{i_{h'}}$ are different for any $h' \le h$ and $i_1, \ldots, i_{h'}$, where $a'_i$, $i \in \{1, \ldots, m-1\}$ is an element in $A'$. Therefore, in the following, we refer to $B_h$ sequences as a nonnegative integer set $A'$ in which the sums of any collection of at most $h$ elements are different.

Let $\mathbb{F}_q$ be a finite field, where $q \ge \max\{(L+1)^h, B\}$ is a prime and $B$ is the number of quantized bins. We construct a $B_h$ sequence $A' \subset \mathbb{F}_q^{2L} = \{a_1, \ldots, a_{2L}\}$ over the field $\mathbb{F}_q$ using the approach in (Bose & Chowla, 1960) (see Appendix G). To encode the data, we label the quantized bins applied on the quantized convex hull of the data with unique elements from $A'$. As before, let $\hat{CH}_+^{(i)}$ and $\hat{CH}_-^{(i)}$ be the quantized convex hulls of the local data at client $i$, and Section 5.2.

Define a label vector $\boldsymbol{v}^{(i)} \in \mathbb{Z}^B$ based on the sets $\hat{CH}_+^{(i)}$ and $\hat{CH}_-^{(i)}$ according to

$$
v_j^{(i)} = \begin{cases} a_{2i}, & \text{if there is a } \hat{\boldsymbol{x}}^{i,j} \in \hat{CH}_+^{(i)} \text{ but no } \hat{\boldsymbol{x}}^{i,j} \in \hat{CH}_-^{(i)} \text{ within the } j\text{th bin,} \\ a_{2i-1}, & \text{if there is a } \hat{\boldsymbol{x}}^{i,j} \in \hat{CH}_-^{(i)} \text{ but no } \hat{\boldsymbol{x}}^{i,j} \in \hat{CH}_+^{(i)} \text{ within the } j\text{th bin,} \\ a_{2i} + a_{2i-1}, & \text{if there is a } \hat{\boldsymbol{x}}^{i,j} \in \hat{CH}_+^{(i)} \text{ and a } \hat{\boldsymbol{x}}^{i,j} \in \hat{CH}_-^{(i)} \text{ within the } j\text{th bin,} \\ 0, & \text{otherwise.} \end{cases}
$$

Note that there is a one-to-one mapping between the vector $\boldsymbol{v}^{(i)}$ and the sets $\hat{CH}_+^{(i)}$ and $\hat{CH}_-^{(i)}$. To calculate $\boldsymbol{v}^{(i)}$ in a federated setting, the clients need an agreement on the client labels (so that the $i$th client uses $a_{2i-1}$ or $a_{2i}$ to label the convex hull). Note that this agreement on the client labels is not revealed to the server. Hence, the server cannot infer the identity of the client that uses labels $a_{2i-1}$ or $a_{2i}$. The agreement can be performed using our decentralized agreement protocol described in Appendix B.

### 5.5 Secure Transmission

To securely communicate the quantized extreme points $\hat{CH}_+^{(i)}$ and $\hat{CH}_-^{(i)}$ at each client $i \in [L]$ to the server, we leverage the SCMA scheme described in (Pan et al., 2023b), where the clients transmit the encoded version of their data such that the server only gets the aggregated data distribution of the union of local

data sets, without leaking information about the identity of individual data. The SCMA scheme encodes data into Reed-Solomon type codes (Reed-Solomon codes are a class of rate-optimal error-correcting codes) and can be shown to be efficient both communication- and computation-wise (Pan et al., 2023b).

Recall that the labels of data class at each server $i$ are described by a vector $\boldsymbol{v}^{(i)}$ based on $B_h$ sequences. The SCMA scheme is used to communicate the vectors $\boldsymbol{v}^{(i)}, i = 1, \ldots, L$. More specifically, each client $i \in \{1, \ldots, L\}$ communicates $(S_1^{(i)}, \ldots, S_{2LK_{max}}^{(i)})$ to the server, where $K_{max} = \max_{i \in [L]}(|\hat{CH}_+^{(i)}| + |\hat{CH}_-^{(i)}|)$ and $S_l^{(i)} = (\sum_{j \in [B]} v_j^{(i)} \cdot j^{l-1} + z_l^{(i)}) \bmod q, i \in [2K_{max}L]$, and $z_l^{(i)}$ is a random key uniformly distributed over the prime field $\mathbb{F}_q$ and hidden from the server. The keys $\{z_l^{(i)}\}_{i \in [L], l \in [2LK_{max}]}$ are generated offline using standard secure model aggregation so that $(\sum_{i \in [L]} z_l^{(i)}) \bmod q = 0$.

To decode, the server uses a Reed-Solomon-type decoder to recover the sum of the label vectors $\boldsymbol{v}^{(i)}$, $i \in [L]$. The server first computes the sum $S_l = (\sum_{i \in [L]} S_l^{(i)}) \bmod q$. Given $S_l$, for $l \in [2LK_{max}]$, the server computes the sum of labels $H_b = \sum_{i \in [L]} v_b^{(i)}$ for each bin index $b \in [B]$ as follows. Since there are at most $K_{max}L$ non-zero entries $H_b$, the server computes the polynomial $g(z) = \prod_{b:H_b \neq 0}(1 - b \cdot z)$ using the Berlekamp-Massey algorithm (Berlekamp, 1968; Massey, 1969). Then, the server factorizes $g(z)$ using the algorithm in (Kedlaya & Umans, 2011), and finds the set $\{b : H_b \neq 0\}$. Finally, the server solves the system of linear equations $S_l = \sum_{l:H_b \neq 0} H_b b^{l-1}$ for $l \in [2K_{max}L]$, by viewing the $H_b$s as unknown variables and $b^{l-1}$, where $H_b \neq 0$, as known coefficients.

Once the values $H_b = \sum_{i \in [L]} v_b^{(i)}$, $b \in [B]$, are obtained, the server retrieves the unique set of indices $i_1, \ldots, i_{h'}$, $h' \leq h$ such that $H_b = a_{i_1} + \ldots + a_{i_{h'}}$, using brute force, which can be done since $a_1, \ldots, a_{2L}$ is a $B_h$ sequence. Finally, the server includes the bin centroid of the $b$th bin into $\hat{CH}_-^{(i)}$ or $\hat{CH}_+^{(i)}$ if $2i - 1 \in \{i_1, \ldots, i_{h'}\}$ or $2i \in \{i_1, \ldots, i_{h'}\}$, respectively. Therefore, the server recovers the convex hulls $\hat{CH}_+^{(i)}$ and $\hat{CH}_-^{(i)}$ for $i \in [L]$ without knowing the exact identity of the clients that contributed the points.

**Remark 5.3.** *In the proposed framework, the server can determine the points in $\hat{CH}_+^{(i)}$ and $\hat{CH}_-^{(i)}$ for $i \in [L]$. While this constitutes data privacy leakage, we emphasize again that the associated client identities for the recovered quantized boundary sets are hidden from the FL server, thus protecting the client identity. Furthermore, these shared convex hull sets contain quantized versions of the original extreme points, further reducing data privacy leakage. Additionally, as observed in Theorem 5.2 and observed from our real-world experiments, the convex hull complexity is small.*

**Communication Complexity.** Using the SCMA transmission protocol from (Pan et al., 2023b), each client $i \in [L]$ communicates $O(K^{(i)}L \max\{h \log L, \log B\})$ bits to the server, where $K^{(i)}$ denotes the total number of extreme points in the quantized convex hulls $\hat{CH}_+^{(i)}$ and $\hat{CH}_-^{(i)}$ (convex hull complexity) of the two local clusters at client $i$, $h$ denotes the number of point collisions (the number of clients that share the same quantized point in the convex hulls of their local clusters), and $B$, as before, equals the total number of quantization bins. Note that the convex hull complexity $K^{(i)}$ tends to be small in practice (which is also corroborated by our experiments on different data sets). In addition, from our experiments we observe that small values of $h$ suffice to avoid collisions and guarantee successful decoding of the local clusters $CH_+^{(i)}$ and $CH_-^{(i)}$, $i \in [L]$, at the server side.

**Computational Complexity.** The encoding algorithm for the convex hulls using the SCMA protocol has complexity $O((K^{(i)})^2 L \log(M_i L))$, where, as before, $M_i$ denotes the number of data points at client $i$.

The decoding of the aggregated labels $\sum_{i \in [L]} \boldsymbol{v}^{(i)}$ in the SCMA scheme has complexity $O(\max_{i \in [L]}[(K^{(i)}L)^{\frac{3}{2}} \log q + K^{(i)}L \log^2 q])$, where $q = \max\{B, h \log L\}$. The brute-force decoding algorithm of the $B_h$ labels of the convex hulls of each cluster has complexity $O(L^h)$.

### 5.6 Balanced Grouping of Convex Hulls at the Server

After the label decoding phase, the FL server can correctly recover the quantized convex hulls $\{\hat{CH}_+^{(i)}\}_{i\in[L]}$ and $\{\hat{CH}_-^{(i)}\}_{i\in[L]}$, albeit without knowing the global ground truth labels of these point sets. The convex hulls have to be grouped correctly to match the ground truth labels. For notational convenience, we denote the set of all quantized convex hulls by $\{CH^{(i)}\}_{i\in[2L]}$, where each $CH^{(i)}$ for $i \in [2L]$ is one of $\{CH_+^{(i)}\}_{i\in[L]}$ or $\{CH_-^{(i)}\}_{i\in[L]}$. We construct a weighted complete undirected graph over $\{CH^{(l)}\}_{l\in[2L]}$ as follows. Let $\mathcal{G}(V)$ be a complete undirected graph with node set $V = \{1, \ldots, 2L\}$, where for simplicity, we use $v \in V$ to represents the convex hull $CH^{(v)}$. For a pair of nodes $u, v \in V$, the edge-weight $w(\{u, v\}) = 1/d_{\mathcal{G}}(u, v)$, where $d_{\mathcal{G}}(u, v)$ is the average pairwise hyperbolic distance between points in the convex hulls $CH^{(u)}$ and $CH^{(v)}$, as described in (8). The grouping of the clusters $\{CH^{(i)}\}_{i\in[2L]}$ is determined by finding the minimum balanced cut on the weighted graph $G$, i.e.,

$$S^* = \underset{S \subset V, |S|=L}{\arg\min} \sum_{u \in S} \sum_{v \in V \setminus S} w(\{u, v\}), \tag{14}$$

for which we use the algorithm in (Kernighan & Lin, 1970). The point sets $PS_1 = \bigcup_{i \in S^*} CH^{(i)}$ and $PS_2 = \bigcup_{i \in [2L] \setminus S^*} CH^{(i)}$ are treated as the two global data point sets, and are assigned, without loss of generality, the global ground truth labels $+1$ and $-1$.

The grouping method for the convex hulls of different clusters by their global labels has complexity $O(L^2 \max_{i_1, i_2 \in [L], i_1 \neq i_2} K^{(i_1)} K^{(i_2)})$ for the graph $G$ construction and complexity $O(IL^2 \log L)$ for computing the balanced minimum cut on $G$, where $I$ denotes the number of iterations of the partitioning procedure (Kernighan & Lin, 1970). We note that due to distortions introduced in the local convex hulls due to quantization, and the heuristic graph partitioning procedure used in obtaining the global clusters, provable guarantees for the correctness of the grouping procedure are not easy to derive. Nevertheless, our proposed solution works well in practice, as demonstrated by our simulation studies of SVM classifier, Section 6.

### 5.7 Learning the SVM Classifier at the Server

As the final step, the server learns a hyperbolic SVM classifier over the labelled global point sets $PS_1$ and $PS_2$, using the procedure described in Section 3. The global reference point $\boldsymbol{p}$ is computed as the geodesic mid-point of the closest pair of extreme points in the convex hulls. The normal vector $\boldsymbol{w}$ is obtained by solving the optimization problem in (6) or (7), as applicable.

We remark that we primarily focus on the linear SVM model. While numerous non-linear kernel SVM models have been thoroughly investigated for Euclidean classification tasks, and adapting the centralized hyperbolic SVM algorithm (Chien et al., 2021; Pan et al., 2023a) to accommodate non-linear kernels is straightforward, since we have access to all data points; in the federated setting, the problem becomes significantly more complex. Here, the server is restricted from directly accessing individual data points. Instead, the server only receives aggregated local convex hulls from clients. While kernel SVM remains feasible at the server, relying solely on convex hulls might compromise the performance of the kernel SVM algorithms. Therefore, effectively integrating non-linear kernels within our current framework that relies on convex hull approaches may be challenging. On the other hand, if one could dispose of the use of convex hulls, then potentially non-linear kernels could be potentially be used, but then we would not know how to resolve the underlying data privacy issues. We believe that the combination of convex hulls and SVM represents the best current option in terms of the trade-off between privacy and the model utility.

## 6 Experiments

We present in what follows results from our numerical studies involving multiple data sets. First, we describe the baseline algorithms used for comparative simulations. Then, we consider binary classification with synthetic data sets that are generated using a newly proposed procedure for sampling uniformly at random from the Poincaré hyperbolic disc. The most important part of the study is to demonstrate how our proposed framework can be applied for multi-label classification tasks on three real-world biological data

sets. The procedure for extending our proposed framework to multi-label scenario is described in Appendix H. Additionally, we provide insights into the impact of quantization bin size on classification accuracy and convex hull complexity. Due to space constraints, some of the details are delegated to Appendix I.

### 6.1 Baselines

We consider the following centralized schemes for comparative simulations. (a) Centralized Poincaré (CP) SVM: We train the linear Poincaré SVM classifier 3 from (Pan et al., 2023a) in a centralized way; (b) Centralized Euclidean (CE): We train the SVM classifier in a centralized way. We also consider the following federated classification baselines: (c) Federated Poincaré (FLP): We use our method from Section 4; (d) Federated Euclidean (FLE): We perform at the server side Euclidean SVM classification over the aggregated global convex hulls.

### 6.2 Synthetic Data Sets

**Data generation.** Our proposed pipeline comprises three main steps. First, $N$ data points are obtained using our proposed Poincaré (radius $R$) uniform sampling procedure described in Alg. 3 of Appendix F. Second, a separating hyperplane is constructed by first sampling a reference point $\boldsymbol{p}$ uniformly at random at a Euclidean distance of $\|\boldsymbol{p}\| = \mu R$ from the origin, and then sampling the normal vector $\boldsymbol{w}$. Here, $0 < \mu < 1$ denotes a scaling parameter. Any point that is within a given hyperbolic distance of $\gamma > 0$ from the hyperplane is removed from the data set. Third, points are labeled as positive (+) or negative (-), depending on which side of the sampled hyperplane they are on. One such example synthetic data set is shown in Figure 4.

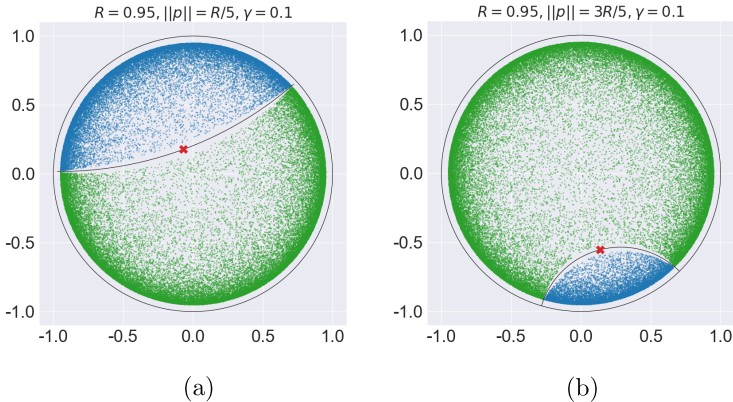

(a)                      (b)

Figure 4: Synthetic data classes constructed as described in the Data generation subsection. The red point denotes the sampled reference point $\boldsymbol{p}$. The geodesic through the red point is the ground truth hyperplane that corresponds to the sampled normal vector. In (a), we set $N = 20,000$, while in (b), we set $N = 60,000$.

**Results.** The results are shown in Figure 5. In the first row, we consider the impact of the choice of reference point used to generate the synthetic data on classification accuracy. As $\|\boldsymbol{p}\|$ increases, i.e., as the reference point is further away from the origin, the ground truth separating hyperplane is more curved in the Euclidean sense (see the ground truth hyperplanes in Figure 4 for comparison). Hence, by increasing $\|\boldsymbol{p}\|$, the performance of both CE and FLE degrades, while the Poincaré baselines achieve near 100% accuracy consistently. In the second row, we report the impact of varying the margin parameter $\gamma$ on classification accuracy. As the margin increases, it becomes easier to learn a Euclidean classifier as the data points corresponding to different labels gradually become linearly separable in the Euclidean sense as well. Hence, the performance of CE as well as FLE improves with increasing $\gamma$. Furthermore, we observe that the federated baselines are almost as good as their centralized counterparts, both for the settings in the first and the second row, demonstrating the effectiveness of our global hulls aggregation method, and supporting the intuition that for SVM classification, extreme points are highly informative. Finally, in the third row, we report the impact of the quantization parameter $\epsilon$ on classification accuracy. As expected, when $\epsilon$ is

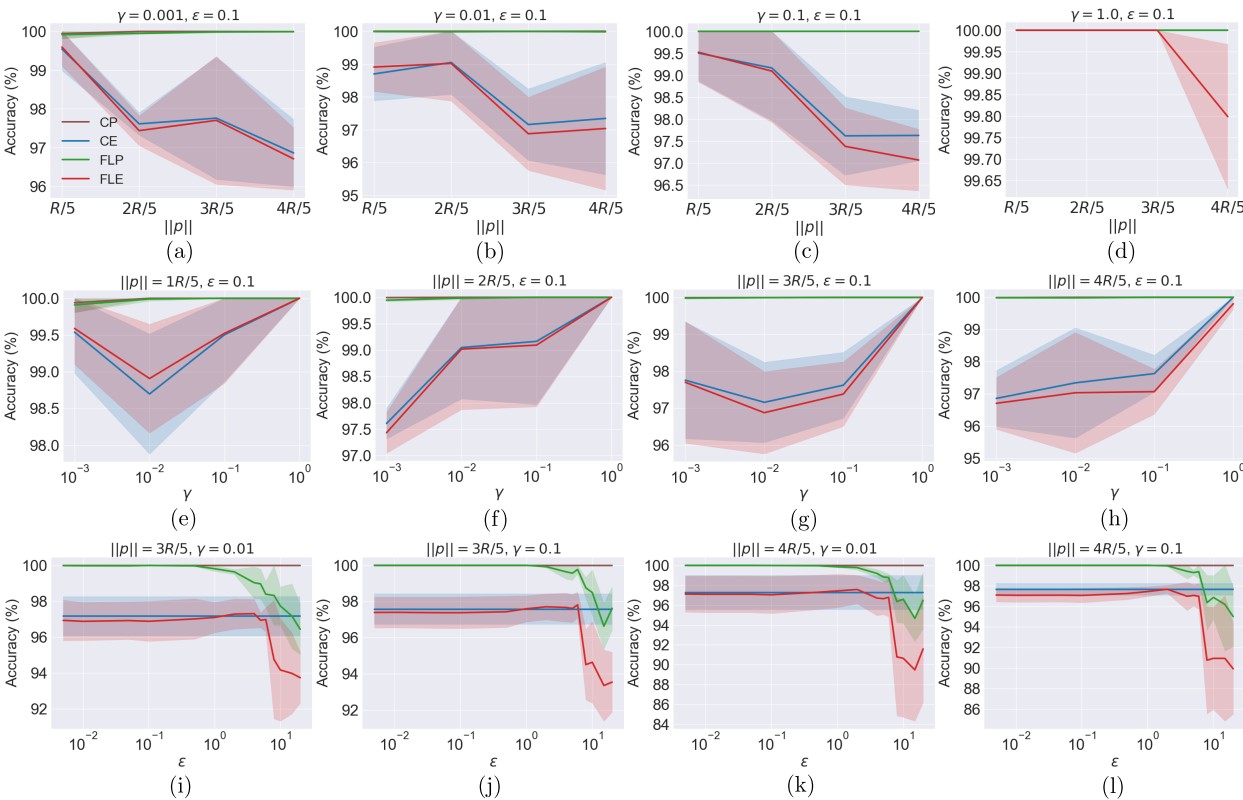

Figure 5: Classification accuracy results for the synthetic data sets. The shaded areas represent the 95% confidence interval for 10 independent trials. (a)-(d) Influence of the parameter $\|\boldsymbol{p}\|$ on the classification accuracy. (e)-(h) Influence of the margin parameter $\gamma$ on the classification accuracy. (i)-(l) Influence of the Poincaré quantization parameter $\epsilon$ on classification accuracy.

small, the quantization regions are small, and hence the local quantized convex hulls that are sent from each client closely approximate the true local convex hulls, resulting in low quantization distortion and no impact on the utility. As $\epsilon$ increases, the increased distortion of the convex hulls affects the classification accuracy significantly, for both FLE and FLP schemes. In particular, the overlap between the quantized convex hulls corresponding to different labels increases as $\epsilon$ increases, making it difficult for the server to learn an SVM classifier, even in the FLP mode.

Table 1: Results for biological data sets, including mean accuracy (%) and 95% confidence interval of the baselines for different data sets. The results are based on 10 independent trials for each setting. The FL method with better mean accuracy is plotted in bold-faced letters.

| Data set | Labels | CP (%) | CE (%) | FLP (%) | FLE (%) |
|---|---|---|---|---|---|
| Olsson | 0-7 | $79.17 \pm 0.00$ | $68.75 \pm 0.00$ | $\mathbf{86.04 \pm 1.41}$ | $75.00 \pm 3.51$ |
| Lung-Human | 0-4 | $72.11 \pm 0.00$ | $65.99 \pm 0.00$ | $\mathbf{69.52 \pm 2.82}$ | $63.54 \pm 2.24$ |
| | $[0, 1, 2, 3]$ | $61.94 \pm 0.00$ | $61.94 \pm 0.00$ | $\mathbf{61.42 \pm 3.19}$ | $60.22 \pm 3.34$ |
| | $[0, 1, 2, 4]$ | $78.52 \pm 0.00$ | $70.37 \pm 0.00$ | $\mathbf{74.15 \pm 5.23}$ | $67.19 \pm 5.98$ |
| | $[0, 1, 3, 4]$ | $70.99 \pm 0.00$ | $61.07 \pm 0.00$ | $\mathbf{66.87 \pm 8.42}$ | $60.99 \pm 6.96$ |
| | $[0, 2, 3, 4]$ | $73.33 \pm 0.00$ | $67.41 \pm 0.00$ | $\mathbf{75.19 \pm 1.44}$ | $71.70 \pm 3.86$ |
| | $[1, 2, 3, 4]$ | $69.64 \pm 0.00$ | $66.07 \pm 0.00$ | $\mathbf{65.00 \pm 2.10}$ | $64.11 \pm 3.73$ |
| UC-Stromal | 0-3 | $73.20 \pm 0.00$ | $73.54 \pm 0.00$ | $\mathbf{70.23 \pm 4.88}$ | $68.21 \pm 4.83$ |
| | $[0, 1, 2]$ | $74.65 \pm 0.00$ | $73.94 \pm 0.00$ | $63.96 \pm 4.66$ | $\mathbf{64.24 \pm 4.92}$ |
| | $[0, 1, 3]$ | $88.47 \pm 0.00$ | $87.39 \pm 0.00$ | $\mathbf{87.77 \pm 1.51}$ | $87.57 \pm 1.24$ |
| | $[0, 2, 3]$ | $80.64 \pm 0.00$ | $82.29 \pm 0.00$ | $\mathbf{79.67 \pm 2.64}$ | $77.86 \pm 3.41$ |
| | $[1, 2, 3]$ | $80.33 \pm 0.00$ | $80.33 \pm 0.00$ | $\mathbf{79.71 \pm 4.80}$ | $79.46 \pm 5.18$ |

### 6.3 Biological Data Sets

**Data sets.** We consider multi-label SVM classification for three biological data sets: Olsson's scRNA-seq data set (Olsson et al., 2016), UC-Stromal data set (Smillie et al., 2019), and Lung-Human data set (Vieira Braga et al., 2019). Our simulations are performed on the Poincaré embeddings of these data sets, which can be obtained using methods described in (Klimovskaia et al., 2020; Skopek et al., 2020). We illustrate the embeddings in Figure 8, which is also included in Appendix I. For simulating the FL multi-label classification tasks, we use subsets of data corresponding to different combinations of labels, for both the UC-Stromal and the Lung-Human data sets. Since the Olsson data set is quite small (it contains only 319 points from 8 classes), we consider the entire data set for multi-label classification. Detailed information about the data sets and experimental settings is available in Appendix I.

**Results.** We present our results in Table 1, which reports mean accuracy (%) and the 95% confidence interval over 10 independent trials for all baseline settings. We observe that FLP almost always outperforms FLE up to $\sim 11\%$, demonstrating that learning a hyperbolic SVM classifier is preferred to simply learning a Euclidean SVM classifier. However, for UC-Stromal (Labels: $[0, 1, 2]$) data, FLE performs slightly better than FLP. This may be attributed to the distortion of the convex hull or particular selection of the reference point. We also observe that for the Olsson (Labels: $0-7$) and Lung-Human (Labels: $[0, 2, 3, 4]$) data sets, the federated baselines have better mean accuracy than their centralized counterparts [1]. Furthermore, the performance of the federated baselines demonstrates that our convex hulls aggregation and label resolution methods indeed perform well in practice, although some of the steps do not have analytical performance guarantees. Finally, we note that for UC-Stromal (Labels: $[0, 1, 2]$) data, federated baselines have a large gap in accuracy compared to their centralized counterparts. This is because the points lying near the decision boundaries of SVM classifiers are not well represented through the local quantized convex hulls.

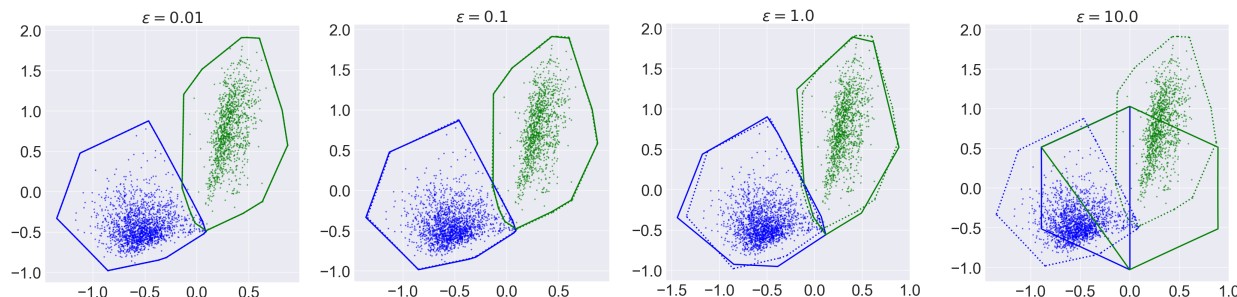

Figure 6: Analysis of the impact of the quantization parameter $\epsilon$ on the convex hull distortion and complexity. The blue and green points correspond to labels 0 and 3 in the UC-Stromal data set. The solid lines denote the quantized convex hulls, while the dotted lines denote the actual convex hulls without quantization.

**Choosing the quantization parameter $\epsilon$.** We use UC-Stromal data set as an example to illustrate the impact of the choice of $\epsilon$ on the performance of the global classifier. In Figure 6, we show how quantization affects the shape of the quantized convex hulls by considering all the data points corresponding to labels 0 and 3. As expected, when $\epsilon$ increases, the shapes of the quantized convex hulls are increasingly distorted with respect to the original convex hulls. To more precisely examine the impact of quantization, we consider our prior experimental setup for labels $0-3$ and analyze how the accuracy and convex hull complexity changes with $\epsilon$. The results are shown in Figure 7. In (a), we observe that federated schemes can suffer a drop in performance when $\epsilon$ increases above 0.1. In (b), we consider the complexity of the quantized convex hulls at the clients corresponding to all the four labels, and plot the average and maximum convex hull complexities across all clients and labels. As expected, the convex hull complexity decreases when increasing $\epsilon$. In (c), we present the ratio of the convex hull complexity and the class size across clients and labels for different choices of the quantization parameter $\epsilon$.

---

[1]We find that training SVM classifiers on extreme points of the convex hulls of the classes instead of the entire classes improves the performance of centralized training, both for Euclidean and Poincaré methods. We believe this to be due to quantization, which may act like a denoising step and improve the performance compared to centralized benchmarks. The results are included in Appendix I.

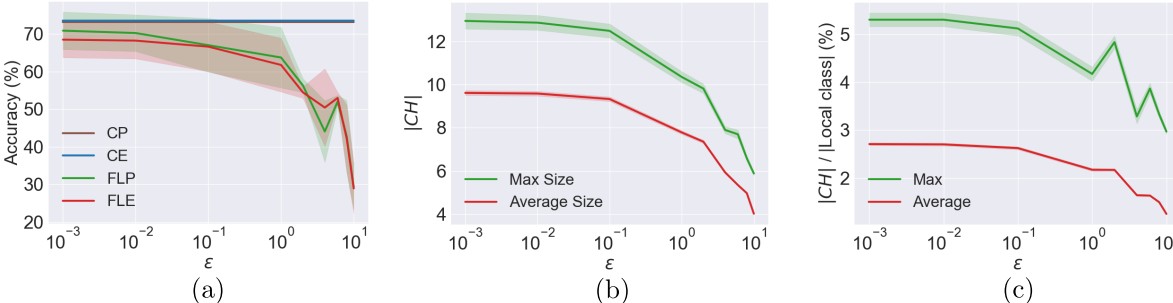

Figure 7: Impact of quantization parameter $\epsilon$ on the accuracy and convex hull complexity of the UC-Stromal data set for the multi-label classification setting with labels $0-3$). Here, as before, $CH$ denotes the quantized convex hull of a client. Average as well as maximum complexities are calculated over all clients and across all local quantized convex hulls. The shaded areas represent the 95% confidence interval from 10 independent trials.

# 7    Conclusion

We introduced a novel approach to federated learning in hyperbolic spaces, thereby addressing challenges in processing hierarchical and tree-like data in distributed and privacy-preserving settings. Specifically, we developed an end-to-end framework for federated learning of SVM classifiers in the Poincaré disc. The key idea behind the approach it to leverage securely aggregated convex hulls for information transfer from the clients to the server. The complexity of convex hulls is analyzed to assess data leakage, and a simple quantization method is proposed for efficient data communication. We also considered detrimental label switching issues and resolved them with a new method based on number-theoretic and coding-theoretic ideas. Additionally, we introduced a novel approach for aggregating client convex hulls using balanced graph partitioning. Experimental results on multiple single-cell RNA-seq data show improved classification accuracy compared to Euclidean counterparts, establishing the utility of privacy-preserving learning in hyperbolic spaces.

Our work also introduced many potentially important new research questions. While information sharing via quantized convex hulls is efficient and secure, extreme points of the quantized convex hulls can contain important identifiable information about outliers, and addressing this limitation is an important problem for future work. Characterizing the impact of quantization on privacy leakage and generalizing the privacy-preserving and communication efficient federated learning protocols to other ML tasks are other questions of interest.

**Acknowledgments**

This work was supported by NSF CIF grant 1956384 and the CZI grant DAF2022-249217.

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

# A    Poincaré Graham Scan

## A.1    Proof of Correctness

**Theorem A.1.** *Given a set of $N$ points $\mathcal{D}$ in the Poincaré disc, Alg. 1 returns $CH(\mathcal{D})$ in $O(N \log N)$ time.*

The proof follows the steps of the proof of the Graham Scan algorithm for Euclidean spaces. However, one has to adapt several steps, such as the initial sorting procedure to accommodate the Poincaré disc as presented in Alg. 1. Before we proceed with the proof, we need the following lemmas. The results use the same definitions and notation as stated in Alg. 1.

**Lemma A.2.** *Let $l_{\boldsymbol{b}}$ be the geodesic starting at $\boldsymbol{b}$ with tangent vector $\boldsymbol{t}$. All points in $\mathcal{D}$ and the origin $\boldsymbol{o}$ lie on the same side of $l_{\boldsymbol{b}}$.*

*Proof.* Since $\boldsymbol{b}$ is the point of largest norm (at the largest distance from the origin), all points in $\mathcal{D}$ and the origin $\boldsymbol{o}$ lie on or inside the circle centered at $\boldsymbol{o}$ and of radius $\|\boldsymbol{o} - \boldsymbol{b}\|_2$. Denote this circle by $O$. By the definition of a geodesic in the Poincaré disc, it is either a circle arc orthogonal to the boundary of the disc or a Euclidean line going through $\boldsymbol{o}$. Clearly, for $\boldsymbol{b} \neq \boldsymbol{o}$, $l_{\boldsymbol{b}}$ is a geodesic of the first type. Also, since $l_{\boldsymbol{b}}$ starts at $\boldsymbol{b}$ by definition, $\boldsymbol{b}$ is the midpoint of the circle arc. As a result, $l_{\boldsymbol{b}}$ is tangent to $O$ and only intersects it at $\boldsymbol{b}$. Therefore, all points in $\mathcal{D}$ and the origin $\boldsymbol{o}$ lie on the same side of $l_{\boldsymbol{b}}$.    □

**Lemma A.3.** *Let $\boldsymbol{a}_1, \boldsymbol{a}_2, \boldsymbol{a}_3$ be points in the Poincaré disc forming a triangle $\Delta \boldsymbol{a}_1 \boldsymbol{a}_2 \boldsymbol{a}_3$ (i.e., not all three points lie on the same geodesic). Then, $sign(CCW(\boldsymbol{a}_1, \boldsymbol{a}_2, \boldsymbol{x})) = sign(CCW(\boldsymbol{a}_2, \boldsymbol{a}_3, \boldsymbol{x})) = sign(CCW(\boldsymbol{a}_3, \boldsymbol{a}_1, \boldsymbol{x}))$ if and only if there exists another point $\boldsymbol{x} \in \Delta \boldsymbol{a}_1 \boldsymbol{a}_2 \boldsymbol{a}_3$.*

*Proof.* Note that a point $\boldsymbol{x} \in \Delta \boldsymbol{a}_1 \boldsymbol{a}_2 \boldsymbol{a}_3$ if and only if it is "on the same side" (i.e. clockwise or counter-clockwise) with respect to all three geodesics $l_{\boldsymbol{a}_1 \to \boldsymbol{a}_2}$, $l_{\boldsymbol{a}_2 \to \boldsymbol{a}_3}$ and $l_{\boldsymbol{a}_3 \to \boldsymbol{a}_1}$. This is equivalent to the condition $\text{sign}(CCW(\boldsymbol{a}_1, \boldsymbol{a}_2, \boldsymbol{x})) = \text{sign}(CCW(\boldsymbol{a}_2, \boldsymbol{a}_3, \boldsymbol{x})) = \text{sign}(CCW(\boldsymbol{a}_3, \boldsymbol{a}_1, \boldsymbol{x}))$, as claimed.    □

We are now ready to prove the correctness of Alg. 1.

*Proof.* By Lemma A.2 all points in $\mathcal{D}$, as well as the origin $\boldsymbol{o}$ are on the same side of the geodesic $l_{\boldsymbol{b}}$. As a result, they are in the same half-plane in the tangent space $\mathcal{T}_{\boldsymbol{b}}$ with a boundary (which is a straight line) corresponding to $l_{\boldsymbol{b}}$. By definition, $\boldsymbol{n}$ is the normal vector of that boundary pointing outward (i.e., towards the side that does not contain any data points). Therefore, the principal angles between each of the $\log_{\boldsymbol{b}}(\boldsymbol{x}_j)$ and $\boldsymbol{t}$ lie in $[0, \pi]$.

Next, we show that retaining only $\mathcal{V} \cup \{\boldsymbol{b}\}$ is correct during the initial sorting step (i.e., showing that $CH(\mathcal{D}) = CH(\mathcal{V} \cup \{\boldsymbol{b}\})$). This is true due to the following reasons. First, note that $\boldsymbol{b}$ must be in the convex hull $CH(\mathcal{D})$. Second, let us assume that $\boldsymbol{c}$ is another point in $CH(\mathcal{D})$ but it is not in $\mathcal{V}$. Based on our initial sorting step, this can only happen when there is some $\boldsymbol{v} \in \mathcal{V}$ such that $\boldsymbol{c}$ is on $l_{\boldsymbol{b} \to \boldsymbol{v}}$. That is, the principal angles for $\log_{\boldsymbol{b}}(\boldsymbol{v}), \log_{\boldsymbol{b}}(\boldsymbol{c})$ are the same with respect to $\boldsymbol{t}$ but $\|\log_{\boldsymbol{b}}(\boldsymbol{v})\| > \|\log_{\boldsymbol{b}}(\boldsymbol{c})\|$. Now there

are two possible scenarios: 1) $\boldsymbol{v} \in CH(\mathcal{D})$; 2) $\boldsymbol{v} \notin CH(\mathcal{D})$. For case 1), it is clear that none of the points on $l_{\boldsymbol{b} \to \boldsymbol{v}}$ belong to $CH(\mathcal{D})$, as $l_{\boldsymbol{b} \to \boldsymbol{v}}$ either lies inside $CH(\mathcal{D})$ or is the boundary of $CH(\mathcal{D})$. In either case, $\boldsymbol{c} \notin CH(\mathcal{D})$. For case 2), $\boldsymbol{v} \notin CH(\mathcal{D})$, which means that $\boldsymbol{v}$ is strictly contained within the convex hull $CH(\mathcal{D})$. Thus, all points on $l_{\boldsymbol{b} \to \boldsymbol{v}}$ (excluding the starting point $\boldsymbol{b}$) are also strictly contained in the convex hull $CH(\mathcal{D})$. As a result, we have established by contradiction that $\boldsymbol{c} \notin CH(\mathcal{D})$ if $\boldsymbol{c} \notin \mathcal{V}$. This immediately implies $CH(\mathcal{D}) = CH(\mathcal{V} \cup \{\boldsymbol{b}\})$.

Next, we focus on showing that the output of Alg. 1 is indeed $CH(\mathcal{V} \cup \{\boldsymbol{b}\})$ and thus is equal to $CH(\mathcal{D})$. We prove this result similarly to what is done for the Graham scan in Euclidean space – via induction. Without loss of generality, let $\mathcal{V} = \boldsymbol{v}_1, \boldsymbol{v}_2, \cdots, \boldsymbol{v}_m$. Let $\mathcal{C}_j = CH(\mathcal{V}[: j] \cup \{\boldsymbol{b}\})$ (i.e., the convex hull of the first $j$ points of $\mathcal{V}$ and $\boldsymbol{b}$). The induction hypothesis is as follows. After the $j$th iteration of the for loop in Alg. 1, points in STACK are points of $\mathcal{C}_j$ in a counter-clockwise order. The base case can be established easily, as $\mathcal{C}_2$ contains points of $CH(\{\boldsymbol{b}, \boldsymbol{v}_1, \boldsymbol{v}_2\}) = \{\boldsymbol{b}, \boldsymbol{v}_1, \boldsymbol{v}_2\}$ (a triangle). Since we start at $\boldsymbol{b}$ and sort $\mathcal{V}$ according to the principal angle (counter-clockwise), the claim is obviously true. For the inductive step, assume the hypothesis holds for $j - 1$. Since the principal angle of $\log_{\boldsymbol{b}}(\boldsymbol{v}_j)$ is greater than $\log_{\boldsymbol{b}}(\boldsymbol{v}_{j-1})$ due to the initial sorting procedure, $\Delta \boldsymbol{b} \boldsymbol{v}_{j-1} \boldsymbol{v}_j$ is not contained in $\mathcal{C}_{j-1}$. Thus, $\boldsymbol{v}_j \in \mathcal{C}_j$. Now, let $\boldsymbol{a}, \boldsymbol{c}$ be the two topmost points of STACK. If CCW$(\boldsymbol{c}, \boldsymbol{a}, \boldsymbol{v}_j) > 0$ or equivalently CCW$(\boldsymbol{c}, \boldsymbol{v}_j, \boldsymbol{a}) < 0$, the algorithm will not pop $\boldsymbol{a}$. This is correct since due to the initial sorting, we know that CCW$(\boldsymbol{b}, \boldsymbol{c}, \boldsymbol{a}) > 0$ and CCW$(\boldsymbol{b}, \boldsymbol{a}, \boldsymbol{v}_j) > 0$ and thus CCW$(\boldsymbol{v}_j, \boldsymbol{b}, \boldsymbol{a}) > 0$. By Lemma A.3, we know that $\boldsymbol{a} \notin \Delta \boldsymbol{b} \boldsymbol{c} \boldsymbol{v}_j$ and thus $\boldsymbol{a} \in \mathcal{C}_j$. In this case, all points in $\mathcal{C}_{j-1}$ also belong to $\mathcal{C}_j$ by the same argument (since their principal angles are sorted). For the case that CCW$(\boldsymbol{c}, \boldsymbol{a}, \boldsymbol{v}_j) \leq 0$, again by Lemma A.3 we know that $\boldsymbol{a} \in \Delta \boldsymbol{b} \boldsymbol{c} \boldsymbol{v}_j$ and thus $\boldsymbol{a} \notin \mathcal{C}_j$. As a result, it is correct to pop out $\boldsymbol{a}$. Note that the while loop will correctly keep outputting points. Assume that the while loop stops popping at $\boldsymbol{v}_j$. Then by the same argument, we know that it is correct to include the entire set $\mathcal{C}_{j-1}$ in $\mathcal{C}_j$. Thus our hypothesis is true. Finally, by setting $j = m$ we arrive the conclusion that the output of Alg. 1 is $\mathcal{C}_m = CH(\mathcal{V} \cup \{\boldsymbol{b}\}) = CH(\mathcal{D})$. This completes the proof. □

Finally, we prove that Alg. 1 produces a minimal convex hull.

*Proof.* Suppose we obtained the (possibly nonminimal) convex hull $\mathcal{C} = \boldsymbol{b}, \boldsymbol{c}_1, \boldsymbol{c}_2, \cdots, \boldsymbol{c}_n$. Note that due to the initial sorting, we know that the "principal angles" of $\boldsymbol{c}_j$ with respect to $\boldsymbol{b}$ are listed in increasing order and that they lie in $[0, \pi]$. As a result, in each Graham Scan check-step (line 10), $\boldsymbol{c}_{j+1}$ is outside the triangle of $\Delta \boldsymbol{b} \boldsymbol{c}_j \boldsymbol{c}_{j-1}$. Thus, even if $\boldsymbol{c}_{j+1}, \boldsymbol{c}_j, \boldsymbol{c}_{j-1}$ lie on the same geodesic (i.e., CCW$(\boldsymbol{c}_{j-1}, \boldsymbol{c}_j, \boldsymbol{c}_{j+1}) = 0$), the algorithm will remove $\boldsymbol{c}_j$ (and is allowed to do so). The only two additional cases to consider are for $\boldsymbol{b}, \boldsymbol{c}_1, \boldsymbol{c}_2$ and $\boldsymbol{c}_{n-1}, \boldsymbol{c}_n, \boldsymbol{b}$ to be on the same geodesic. Due to initial sorting, from a set of points with the same principal angle, we will only keep the furthest one (line 5). Therefore, the Poincaré Graham Scan indeed returns the minimal convex hull. □

## B   Decentralized Order Agreement for $B_h$ Label Assignments

We describe next how the participating clients can agree upon an ordering of themselves while keeping it hidden from the FL server – the ordering has to be protected only from the server, and not the clients themselves as they do not have information about the convex hulls of other clients.

Clients participate in a key exchange protocol, such as the Diffie-Hellman key agreement method, that enables any pair of clients to securely exchange secret information over an insecure communication channel (Diffie & Hellman, 1976; Bonawitz et al., 2017). In essence, the key agreement protocol allows any pair of clients to establish a common secret key that can be used for secure communication, even in the presence of potential eavesdroppers, thus ensuring the confidentiality and integrity of their communication. Particularly, in the first step of the key agreement protocol, each client $i \in [L]$ independently generates a pair of keys: A public key $\mathcal{K}^{(i,PK)}$ and a secret key $\mathcal{K}^{(i,SK)}$. As the names suggest, the public key $\mathcal{K}^{(i,PK)}$ of client $i$ is accessible to other clients and the server, while the private key is not. In the next step of the key agreement protocol, each pair of clients $i_1, i_2 \in [L]$ agrees on a random secret key $\mathcal{R}_\mathcal{K}^{(i_1, i_2)}$ known only to clients $i_1$ and $i_2$, where $\mathcal{R}_\mathcal{K}^{(i_1, i_2)} = \mathcal{R}_\mathcal{K}^{(i_2, i_1)}$ is a function of the public key $\mathcal{K}^{(i_1, PK)}$ and secret key $\mathcal{K}^{(i_2, SK)}$. Thus, at the end of the key agreement process, each pair of clients shares a common secret key, not known to other clients or the server.

For decentralized order agreement, we further assume that clients have access to a public pseudo-random number generator $PRG(\cdot)$. Each client $i$ uses $PRG(\cdot)$ and obtains a private random number $n^{(i)} = PRG(\mathcal{K}^{(i,SK)})$. Thereafter, each pair of clients $i_1$ and $i_2$ secretly exchange their private random numbers $n^{(i_1)}$ and $n^{(i_2)}$ using their common secret key $\mathcal{R_K}^{(i_1,i_2)}$ for encryption/decryption. Notably, this ensures that the private numbers $\{n^{(i)}\}_{i \in [L]}$ remain hidden from the server. Next, the clients sort the list of the random numbers $\{n^{(i)}\}_{i \in [L]}$. Each client $i$ then determines its position in the shared random ordering by finding the position of its number $n^{(i)}$ in the sorted list. Let $\Pi(i)$ denote the corresponding position of client $i$ in the shared random ordering. Then, the client picks the numbers $a_{2i-1} = 2\Pi(i) - 1$ and $a_{2i} = 2\Pi(i)$ from the $B_h$ sequence for labeling of points in their quantized convex hulls $\hat{CH}_+^{(i)}$ and $\hat{CH}_-^{(i)}$, respectively.

## C   Uniform Poincaré Quantization

We describe another approach for Poincaré disc quantization, using ideas that are similar to those described in the Poincaré Quantization, Section 5.2. We once exploit radial symmetry and form the quantization bins by intersecting straight lines passing through the origin and concentric circles around the origin. However, in this new setting, our goal is to quantize the bounded region within the circle of hyperbolic radius $R_H$ so that *each quantization bin has the same area*. With this constraint, we can use finely-grained quantization bins to perform uniformly at random sampling of points on the Poincaré disc.

First, observe that the area of a circle with the hyperbolic radius of $r_H$ is given by the following formula (Hitchman, 2009):

$$A(r_H) = 4\pi s^2 \sinh^2 \left( \frac{r_H}{2s} \right), \tag{15}$$

where $s = 1/\sqrt{k}$ and $k$ is curvature constant. Let $N_\Theta$ denote the number of angular quantization bins. Additionally, let $N_{R_H}$ denote the number of radial quantization bins for a hyperbolic radius $R_H$ and any given angle. The total number of quantization bins therefore equals $B = N_\Theta N_{R_H}$, and the area of each bin is $\frac{A(R_H)}{B}$. Thus, the boundaries for the bins in the angular direction can be denoted by $\phi_{n_1} = \frac{n_1}{N_\Theta} 2\pi$, where $n_1 \in [N_\Theta]$. In what follows, we characterize the radial boundaries $h_{n_2}$ for the quantization bins, where $n_2 \in [N_{R_H}]$ and $h_{N_{R_H}} = R_H$.

Consider the circle with hyperbolic radius $h_{n_2}$. Since each quantization bin has same area in the hyperbolic plane, we have the following relationship between the circles with hyperbolic radii $h_{n_2}$ and $R_H$:

$$A(h_{n_2}) = \frac{n_2}{N_{R_H}} A(R_H), \tag{16}$$

which results in the following relationship between $h_{n_2}$ and $R_H$:

$$h_{n_2} = 2s \sinh^{-1} \left( \sqrt{\frac{n_2}{N_{R_H}}} \sinh \left( \frac{R_H}{2s} \right) \right). \tag{17}$$

Therefore, a quantization bin $B(n_1, n_2)$ is well-defined via its angular boundaries $\phi_{n_1-1}$ and $\phi_{n_1}$, and radial boundaries $h_{n_1-1}$ and $h_{n_1}$, where $\phi_0 = 0$ and $h_0 = 0$. Furthermore, any point in the quantization bin $B(n_1, n_2)$ is mapped to the bin-center $bc(n_1, n_2)$, which we define as the point in the bin which partitions it into four parts of equal area. Based on similar arguments as above, the angular distance $\phi_{bc(n_1,n_2)}$ and radial hyperbolic distance $h_{bc(n_1,n_2)}$ for the bin-center are as follows:

$$\phi_{bc(n_1,n_2)} = \frac{\phi_{n_1-1} + \phi_{n_1}}{2} \quad \text{and} \quad h_{bc(n_1,n_2)} = 2s \sinh^{-1} \left( \sqrt{\frac{n_2 - 0.5}{N_{R_H}}} \sinh \left( \frac{R_H}{2s} \right) \right). \tag{18}$$

To map the results from the hyperbolic plane to the Poincaré disc, the Euclidean radius of the corresponding bin-center is obtained using (9) and its 2D projection is obtained using $\phi_{bc(n_1,n_2)}$.

As an example, consider a point $\boldsymbol{x} \in \mathbb{B}_k^2$ within a bin $B(n_1, n_2)$. It is quantized to

$$\hat{\boldsymbol{x}} = (\alpha \cos(\zeta), \alpha \sin(\zeta)), \tag{19}$$

where $\zeta = \phi_{bc(n_1,n_2)}$, $\alpha = s(e^{\frac{\tau}{s}} - 1)/(e^{\frac{\tau}{s}} + 1)$ and $\tau = h_{bc(n_1,n_2)}$. This approach of uniform area quantization is key to implementing the Poincaré uniform sampling process used in Alg. $3^2$, as well as for proving the convex hull complexity result of Theorem 5.2.

## D  Proof for the Convex Hull Complexity

We show that the expected number of extreme points of the convex hull of $N$ points uniformly sampled from a Poincaré disc is at most $O(N^{\frac{1}{3}})$. The proof follows similar ideas as those used for the convex hull complexity results in Euclidean spaces (Har-Peled, 2011), which are adapted to the hyperbolic setting and involve some technical arguments unique to hyperbolic spaces. We start with a lemma showing that the expected fraction of points that are extreme points of the convex hull is upper bounded by the expected fraction of the area in the Poincaré disc not covered by the convex hull. The lemma appears in (Har-Peled, 2011) for the Euclidean space, and in this setting, it is known as Efron's theorem (Efron, 1965).

**Lemma D.1.** *Let $C$ be a Poincaré disc of radius $R < s$. Let $f(N) \in [0,1]$ be a function of an integer $N \geq 0$ such that the expected area of the convex hull of $N$ points distributed independently and uniformly at random over $C$ is at least $(1 - f(N))Area(C)$. Then, the expected number of extreme points of the convex hull of the $N$ points is at most $Nf(\frac{N}{2})$.*

*Proof.* The proof is the same as that for the Euclidean space (Har-Peled, 2011), and is presented for completeness. Uniformly at random partition the $N$ points into two subsets, $S_1$ and $S_2$, both of cardinality $\frac{N}{2}$. Let $V_1$ and $V_2$ be the sets of extreme points of the convex hull of $S_1 \cup S_2$ that are in $S_1$ and $S_2$, respectively. Denote the convex hull of $S_2$ as $CH(S_2)$.

It follows from definition that the expected number of extreme points of the convex hull of $S_1 \cup S_2$ is $E[|V_1|] + E[|V_2|]$. Moreover, we have that

$$E\left[|V_1|\,\Big|\,S_2\right] \leq \frac{N}{2}\left(\frac{Area(C) - Area(CH(S_2))}{Area(C)}\right),$$

since the probability of a point in $S_1$ lying in $CH(S_2)$, and thus not in $V_1$ (because a point strictly inside the convex hull is not an extreme point) is at least $\frac{Area(CH(S_2))}{Area(C)}$. Then, it follows that

$$
\begin{aligned}
E[|V_1|] &= E_{S_2}\left[E\left[|V_1|\,\Big|\,S_2\right]\right] \\
&\leq \frac{N}{2}\left(\frac{f(\frac{N}{2})Area(C)}{Area(C)}\right) = \frac{N}{2}f(\frac{N}{2}).
\end{aligned}
$$

Similarly, we have that $E[|V_2|] \leq \frac{N}{2}f(\frac{N}{2})$. Therefore, the expected number of extreme points of the convex hull of $S_1 \cup S_2$ is at most $Nf(\frac{N}{2})$. $\qquad\square$

Lemma D.1 implies that if the area of the convex hull of the independently and uniformly at random sampled $N$ points approaches the area of the Poincaré disc, then the expected fraction of the points that are extreme points of the convex hull goes to zero. In what follows, we show that the area of the convex hull is at least $(1 - O(N^{\frac{2}{3}}))$ times the area of the Poincaré disc, i.e., $f(N) \leq O(N^{\frac{2}{3}})$. Then, it follows from Lemma D.1 that the expected convex hull complexity is at most $O(N^{\frac{1}{3}})$.

Partition the Poincaré disc into $m = N^{\frac{1}{3}}$ sectors $T_1, \ldots, T_m$, each having an angle $\frac{2\pi}{m}$. Let $D_1, \ldots, D_{m^2}$ be $m^2$ discs such that $D_1$ is the disc $C$ and the area of $D_l$ minus the area of $D_{l+1}$ is the same for $l \in [m^2]$, where the area of $D_{m^2+1}$ is 0. Let subsector $T_{l,j} = T_l \cap D_j$, $l \in [m], j \in [m^2]$ be the intersection of sector $T_l$ and disc $D_j$. The subsectors $T_{l,j}$ have equal area. This partition is the same as the one for the quantization

---

$^2$All that is needed for uniform sampling is to construct a quantizer with a sufficiently large number of bins, and then select points uniformly at random from a discrete collection of bin labels. The sampled point is the centroid of the bin. For more details, see Section F.

bins described in Section C, where $N_\Theta = m$ and $N_{R_H} = m^2$. We use specific notation for the sectors only in this section since this is required for formal analysis.

For any sector $T_l$, let $X_l \in [1, m^2]$ be the smallest integer such that at least one of the $N$ points lies in the subsector $T_{l,X_l}$. Then $Pr(X_l = t) \leq (1 - \frac{t-1}{N})^N \leq e^{-(t-1)}$, since the probability on the left-hand-side is at most the probability that the subsectors $T_{l,1}, \ldots, T_{l,t-1}$ do not contain any of the $N$ points. Hence, we have

$$E[X_l] = \sum_{l=1}^{m^2} t Pr(X_l = t) = O(1). \tag{20}$$

Let $K_o$ be the convex hull of the union of the $N$ points and the origin $o$. We show that $K_o$ completely covers at least $m^2 - (X_{l+1} + X_{l-1} + O(1))$ subsectors $T_{l,X_{l+1}+X_{l-1}+O(1)}, \ldots, T_{l,m^2}$ in sector $T_l$. Let $P$ and $Q$ be the points that lie in the subsectors $T_{l-1,X_{l-1}}$ and $T_{l+1,X_{l+1}}$, respectively, where $T_{m+1,j} = T_{1,j}$ and $T_{0,j} = T_{m,j}$ for $j \in [1, m^2]$. We show that the convex hull of the origin $o$ and the points $P$ and $Q$ covers at least $m^2 - (X_{l+1} + X_{l-1} + O(1))$ subsectors $T_{l,X_{l+1}+X_{l-1}+O(1)}, \ldots, T_{l,m^2}$ in sector $T_l$.

Without loss of generality, assume that the length of the geodesic $oP$ (which is a straight line in the Poincaré disc) is larger than the geodesic $oQ$. Pick a point $P'$ on $oP$ such that the length of $oP'$ equals the length of $oQ$. Then, $P'$ lies in the subsector $T_{l-1,X_{l+1}}$. Consider the geodesic between $P'$ and $Q$, denoted by $P'Q$. We have the following lemma.

**Lemma D.2.** *The minimum distance from the origin $o$ to a point on the geodesic $P'Q$ is at least $R_{X_{l+1}+X_{l-1}+O(1)}$, where $R_j$, $j \in [1, m^2]$ is the smallest radius of a disc centered at the origin that completely covers $T_{l,j}$ for $l \in [1, m]$.*

By virtue of Lemma D.2, the subsectors $T_{l,j}$, $j \in [X_{l+1} + X_{l-1} + O(1), m^2]$ are covered by the convex hull of $o$, $P'$ and $Q$, and thus covered by $K_o$. Hence, the total number of subsectors $T_{l,j}$ covered by $K_o$ is at least $\sum_{l=1}^{m}[m^2 - (X_{l+1} + X_{l-1} + O(1))]$. This implies that the expected area of $K_o$ is at least

$$\frac{\sum_{l=1}^{m}[m^2 - (E[X_{l+1}] + E[X_{l-1}] + O(1))]}{m^3} = 1 - O(N^{-\frac{2}{3}})$$

times the area of the Poincaré disc. Note that the probability that the origin $o$ lies outside the convex hull of the $N$ points is the probability that there exists a point $x$ among the $N$ points such that the remaining $N-1$ points lie on the same side of the geodesic passing through $o$ and $x$. This probability equals $\frac{N}{2^{N-1}}$. Therefore, we conclude that the expected area of the convex hull of the $N$ points is at least $(1 - Pr(o \in K))(1 - O(N^{-\frac{2}{3}}))Area(C) + Pr(o \notin K)0 = 1 - O(N^{-\frac{2}{3}})Area(C)$, where $C$ is the Poincaré disc and $K$ is the convex hull of the $N$ points.

To complete the proof, we now prove Lemma D.2. By the geometric property of a geodesic on a Poincaré disc, $P'Q$ is part of a circle $o'$ centered at a point $o'$ that is orthogonal to the Poincaré disc centered at $o$ with radius $s$, in the Euclidean space. The distance from $o$ to the geodesic $P'Q$ is the difference between the length of $oo'$ and the radius of the circle $o'$. Let $R'$ be the radius of the circle $o'$.

Let the length of $oP'$ be $r$ and let the line $oP'$ intersect the circle $o'$ at another point $P''$. Let the length of $oP''$ be $r'$. Then, we have that $rr' = s^2$ since the circle $o'$ is orthogonal to the unit Poincaré disc centered at $o$. Let $M$ be the midpoint of the chord $P'P''$. Then $o'M$ is perpendicular to $oM$ and the length of $oM$ is $\frac{r+r'}{2} = r + \frac{s^2}{r}$. Let the angle between $oP'$ and $oQ$ be $\theta_{P'Q}$, which is at most $\frac{6\pi}{m}$. Since the lengths of $oP'$ and $oQ$ are equal, the angle between $oM$ and $oo'$ is $\frac{\theta_{P'Q}}{2}$. Hence, the length of $oo'$ is

$$\frac{r + \frac{s^2}{r}}{2 \cos \frac{\theta_{P'Q}}{2}}.$$

On the other hand, the length of $oo'$ equals $\sqrt{(R')^2 + s^2}$. Therefore, we have

$$\frac{r + \frac{s^2}{r}}{2 \cos \frac{\theta_{P'Q}}{2}} = \sqrt{(R')^2 + s^2}.$$

Then, the distance from $o$ to the geodesic $P'Q$ can be written as

$$\sqrt{(R')^2 + s^2} - R' = r - \frac{(r^2 + s^2)(\theta_{P'Q})^2}{2(\frac{s^2}{r} - r)} + o((\theta_{P'Q})^2) = r - O(\frac{1}{m^2}).$$

It is left to be shown that this distance $r - O(\frac{1}{m^2})$ is at least $R_{X_{l+1} + X_l - 1 + O(1)}$, where $R_l$ is the radius of the disc $D_l$, $l \in [1, m^2]$. Note that by assumption, the length of $oP'$ equals the length of $oQ$. Hence, $r \in [R_{X_{l+1}}, R_{X_{l+1}-1}]$. Moreover, $R_l$ can be obtained from (17) by reversing the indexing order and then from (9), i.e.,

$$h_l = 2s \sinh^{-1}\left(\sqrt{\frac{m^2 + 1 - l}{m^2}} \sinh\left(\frac{R_H}{2s}\right)\right),$$

$$R_l = s\left(\frac{e^{\frac{h_l}{s}} - 1}{e^{\frac{h_l}{s}} + 1}\right), \tag{21}$$

where $R_H = s \ln\left(\frac{s+R}{s-R}\right)$ and $R$ is the radius of the Poincaré disc $C$. Since

$$r - O(\frac{1}{m^2}) \geq R_{X_{l+1}} - O(\frac{1}{m^2})$$

$$= s\left(\frac{e^{\frac{h_{X_{l+1}} - O(\frac{1}{m^2})}{s}} - 1}{e^{\frac{h_{X_{l+1}} - O(\frac{1}{m^2})}{s}} + 1}\right),$$

and

$$\sinh(\frac{h_{X_{l+1}} - O(\frac{1}{m^2})}{2s}) = \sinh(\frac{h_{X_{l+1}}}{2s}) - \cosh(\frac{h_{X_{l+1}}}{2s})O(\frac{1}{m^2})$$

$$\geq \sinh(\frac{h_{X_{l+1} + O(1)}}{2s}),$$

we have $h_{X_{l+1}} - O(\frac{1}{m^2}) \geq h_{X_{l+1} + O(1)} \geq h_{X_l + X_{l+1} + O(1)}$, and thus

$$r - O(\frac{1}{m^2}) \geq s\left(\frac{e^{\frac{h_{X_l + X_{l+1} + O(1)}}{s}} - 1}{e^{\frac{h_{X_l + X_{l+1} + O(1)}}{s}} + 1}\right) = R_{X_l + X_{l+1} + O(1)}.$$

## E  Proof of Theorem 5.2

We now present the expected convex hull complexity when the data points in the Poincaré disc are quantized, following the algorithm in Section 5.2. Specifically, let $\epsilon$ be the distance margin of the quantization algorithm in Section 5.2. We first restate Theorem 5.2 in the following.

**Theorem.** *Let $x_1, \ldots, x_N$ be $N$ points uniformly distributed over a Poincaré disc of radius $R < s$, where $N$ is sufficiently large. Let $\hat{x}_1, \ldots, \hat{x}_N$ be quantized versions of $x_1, \ldots, x_N$, obtained using the quantization rule in (11) and (12), with distance margin $\epsilon = O(N^{-c})$, and some constant $c > 0$. Then, the expected number of extreme points on the minimal convex hull of $\hat{x}_1, \ldots, \hat{x}_N$ is at most $O(N^c)$ when $c < \frac{1}{2}$, at most $O(N^{1-c})$ when $\frac{1}{2} \leq c \leq \frac{2}{3}$, and at most $O(N^{\frac{1}{3}})$ when $\frac{2}{3} < c$.*

Before proving Theorem 5.2, we first state the following lemma, which is a modification of Lemma D.1 for quantized data points.

**Lemma E.1.** *Let $C$ be a Poincaré disc of radius $R < s$ and let $f_1(N) \in [0, 1]$ be a function of an integer $N \geq 0$ such that the expected area of the convex hull of $\hat{x}_1, \ldots, \hat{x}_N$ is at least $(1 - f_1(N))Area(C)$. Then the expected number of extreme points of the convex hull of $\hat{x}_1, \ldots, \hat{x}_N$ is at most $N f_1(\frac{N}{2})$.*

The proof of Lemma E.1 follows along the same lines of that for Lemma D.1.

In what follows, we prove Theorem 5.2.

*Proof.* (of Theorem 5.2) We first prove the theorem for $c < \frac{1}{2}$. Similar to what was done in Appendix D, we partition the Poincaré disc into $N_\Theta = 2Cir(R_H)/\epsilon = O(N^c)$ (See Section 5.2 for definition of $N_\Theta$) sectors $T_1, \ldots, T_{N_\Theta}$ and define $N^c$ discs $D_1, \ldots, D_{N^c}$ such that $D_1$ is the Poincaré disc $C$ and the difference between the area of $D_l$ and the area of $D_{l+1}$ is the same for $l \in [N^c]$. Then we define subsectors $T_{l,j} = T_l \cap D_j$, $l \in [N_\Theta]$, $j \in [N^c]$, of the same area.

Then, the probability that a given subsector $T_{l,j}$ is empty is at most $(1 - \frac{1}{N_\Theta N^c})^N \leq e^{-O(N^{1-2c})}$. By the union bound, the probability that there exists at least one empty subsector is at most $N_\Theta N^c e^{-O(N^{1-2c})}$.

We now show that when every subsector contains at least one point, the number of extreme points of the convex hull of $\hat{\boldsymbol{x}}_1, \ldots, \hat{\boldsymbol{x}}_N$ is $N_\Theta = O(N^c)$. This follows from the fact that in each sector, each subsector $T_{l,1}$, $l \in [N_\Theta]$, that is on the boundary of the Poincaré disc $C$ is contained in the quantization bin given by (11) and (12), that is on the boundary of the Poincaré disc $C$. Moreover, each centroid in the quantization bin that is on the boundary of the Poincaré disc is an extreme point of the convex hull when every quantization bin has at least one data point.

Let $Q$ denote the event that there exists an empty quantization bin and $V$ the number of extreme points of the convex hull of $\hat{\boldsymbol{x}}_1, \ldots, \hat{\boldsymbol{x}}_N$. Then, we have

$$E[V] = Pr(Q)E[V|Q] + Pr(Q^c)E[V|Q^c] = N_\Theta + o(1) = O(N^c).$$

We now consider an upper bound for the case when $\frac{1}{2} \leq c \leq \frac{2}{3}$. We partition the Poincaré disc into $N^{1-c}$ sectors $T_1, \ldots, T_{N^{1-c}}$, and subpartition each sector $T_l$ into $N^c$ subsectors $T_{l,1}, \ldots, T_{l,N^c}$ of equal area, separated by $N^c - 1$ circles. The distance between the origin and the subsector $T_{l,j}$ decreases with $j \in [N^c]$ for every $l \in [N^{1-c}]$. Similar to what was done in Section D, let $X_l$ be the smallest integer such that one of $\boldsymbol{x}_1, \ldots, \boldsymbol{x}_N$ lies in the subsector $T_{l,X_l}$. Let $Y_l$ be the smallest integer such that $\hat{\boldsymbol{x}}_1, \ldots, \hat{\boldsymbol{x}}_N$ lies in the subsector $T_{l,Y_l}$. The difference between the distance between the origin $\boldsymbol{o}$ and $\boldsymbol{x}_l$ and the distance between $\boldsymbol{o}$ and $\hat{\boldsymbol{x}}_l$ is at most $\epsilon$. Hence,

$$R_{Y_l} - \epsilon \leq R_{X_l} \leq R_{Y_l} + \epsilon, \tag{22}$$

where $R_l$, $l \in [N^c]$ is the radius of $D_l$ in the Poincaré disc. Similarly to (21), $R_l$ can be obtained by

$$r_l = 2s \sinh^{-1}\left(\sqrt{\frac{N^c + 1 - l}{N^c}} \sinh\left(\frac{R}{2s}\right)\right),$$

$$R_l = s\left(\frac{e^{\frac{r_l}{s}} - 1}{e^{\frac{r_l}{s}} + 1}\right). \tag{23}$$

From $\epsilon = O(\frac{1}{N^c})$, (23), and (22), we have

$$X_l \geq Y_l - O(1). \tag{24}$$

and

$$R_l - O\left(\frac{1}{N^c}\right) = R_{l+O(1)}. \tag{25}$$

Since $E[X_l] = O(1)$ according to (20), we have $E[Y_l] = O(1)$. Let $\hat{P}$ and $\hat{Q}$ be two points among $\hat{\boldsymbol{x}}_1, \ldots, \hat{\boldsymbol{x}}_N$ that are in the subsector $T_{l-1,Y_{l-1}}$ and $T_{l+1,Y_{l+1}}$, respectively. Following the same argument as in the proof of Lemma D.2, we can establish that the minimum Euclidean distance between the origin $\boldsymbol{o}$ and the geodesic $\hat{P}\hat{Q}$ is at least $R_{Y_{l-1}+Y_{l+1}} - O(\frac{1}{N^{2-2c}}) \geq R_{Y_{l-1}+Y_{l+1}} - O(\frac{1}{N^c}) = R_{Y_{l-1}+Y_{l+1}+O(1)}$. Then following the same

argument as in Section D, we can prove that $f_1$ in Lemma E.1 can be upper bounded by $O(\frac{1}{N^c})$, which by Lemma E.1 implies that the expected number of extreme points on the convex hull of $\hat{\boldsymbol{x}}_1, \ldots, \hat{\boldsymbol{x}}_N$ is at most $O(N^{1-c})$.

The proof of the upper bound for the case $c > \frac{2}{3}$ is similar to that for the case $\frac{1}{2} \leq c \leq \frac{2}{3}$, where instead of partitioning the Poincaré disc into $N^{1-c}$ equal-sized sectors and partitioning each sector into $N^c$ subsectors, we equally partition the Poincaré disc into $N^{\frac{1}{3}}$ sectors and partition each sector into $N^{\frac{2}{3}}$ subsectors. The remaining steps are similar. $\qquad\square$

## F  Poincaré Uniform Sampling

Our sampling method is based on the intuition that the probability of sampling a point from a region in the Poincaré disc should be proportional to the hyperbolic area of the region. Based on this idea, we adapt the sampling procedure from our proposed Poincaré uniform quantization method in Appendix C by taking the limit of both the number of angular quantization bins ($N_\Theta$) and the number of radial quantization bins ($N_{R_H}$) to infinity. The sampling procedure is described in Alg. 3.

---
**Algorithm 3** Poincaré Uniform Sampling

1: **Input:** number of points $N$, bound on the Poincaré radius $R$, constant curvature $k$.
2: Compute $s = 1/\sqrt{k}$, $R_H = s \ln\left(\frac{s+R}{s-R}\right)$.
3: Sample $N$ iid points $\eta_1, \ldots, \eta_N$ from the uniform $\mathcal{U}(0,1]$ distribution.
4: Sample $N$ iid points $\zeta_1, \ldots, \zeta_N$ from the uniform $\mathcal{U}(0,2\pi]$ distribution.
5: Compute $\tau_j = 2s \sinh^{-1}\left(\sqrt{\eta_j} \sinh\left(\frac{R_H}{2s}\right)\right)$ $\forall j \in [N]$.
6: Compute $\alpha_j = s(e^{\frac{\tau_j}{s}} - 1)/(e^{\frac{\tau_j}{s}} + 1)$ $\forall j \in [N]$.
7: Obtain sample $\boldsymbol{x}_j \in \mathbb{B}_k^2$ by computing $\boldsymbol{x}_j = (\alpha_j \cos\zeta_j, \alpha_j \sin\zeta_j)$ $\forall j \in [N]$.
8: **Return** point set $\{\boldsymbol{x}_j\}_{j \in [N]}$.

---

## G  Construction of $B_h$ Sequences

We now describe how to construct a $B_h$ sequence used for label switching resolution. We start with the construction (with proof of correctness) in (Bose & Chowla, 1960) that presents a set of numbers $a'_1, \ldots, a'_{m+1}$ such that: (1) $m + 1 = p^\ell$, where $p$ is a prime; (2) $a'_1, \ldots, a'_{m+1} \in \{0, 1, \ldots, (m+1)^h - 1\}$; (3) for all $1 \leq i_1 \leq \ldots \leq i_h \leq m + 1$, the sums $a'_{i_1} + a'_{i_2} + \ldots + a'_{i_h}$ are distinct.

Let $\alpha_1 = 0, \alpha_2, \ldots, \alpha_m$ be the elements of a Galois field $\mathcal{F}_{m+1}$. Let $\beta$ be a primitive root of the extension field $\mathcal{F}_{(m+1)^h}$, which implies that $\beta^0, \beta^1, \ldots, \beta^{(m+1)^h - 1}$ constitute all the nonzero elements in $\mathcal{F}_{(m+1)^h}$ and that $\beta$ is not the root of any irreducible polynomial over the ground field of degree smaller than $h$.

Let
$$\beta^{a'_l} = \beta + \alpha_l, \ l \in [m+1], a'_l \leq (m+1)^h - 1.$$

We show next that the sums $a'_{l_1} + a'_{l_2} + \ldots + a'_{l_h}$ are all distinct, for $1 \leq l_1 \leq \ldots \leq l_h \leq m + 1$. Suppose on the contrary that
$$a'_{l_1} + a'_{l_2} + \ldots + a'_{l_h} = a'_{j_1} + a'_{j_2} + \ldots + a'_{j_h}.$$

Then, by definition, we have
$$\beta^{a'_{l_1}} \beta^{a'_{l_2}} \ldots \beta^{a'_{l_h}} = \beta^{a'_{j_1}} \beta^{a'_{j_2}} \ldots \beta^{a'_{j_h}},$$

which implies that
$$(\beta + \alpha_{l_1})(\beta + \alpha_{l_2}) \ldots (\beta + \alpha_{l_h}) = (\beta + \alpha_{j_1})(\beta + \alpha_{j_2}) \ldots (\beta + \alpha_{j_h}).$$

Then $\beta$ is the root of a polynomial with degree less than $h$ and coefficients from the ground field, contradicting the fact that $\beta$ is a primitive element of $\mathcal{F}_{(m+1)^h}$. Therefore, the sums of $h$ elements in $a'_1, \ldots, a'_h$ are all distinct.

Finally, as described in Section 5.4, we can construct a $B_h$ sequence set with $m$ elements by simply taking the differences $a_l = a'_{l+1} - a'_l$.

## H   Extension to Multi-Label Classification

We now describe how our proposed framework extends to settings with $J > 2$ global ground truth labels. The process of computing the quantized convex hull for each local cluster remains same. For secure communication of quantized convex hulls, each client simply needs $J$ unique numbers from the $B_h$ sequence instead of just 2, while all other system components remain unaffected. For graph based grouping, the FL server obtains the representative graph with $JL$ nodes and partitions it into $J$ groups of nodes using spectral clustering (Shi & Malik, 2000; Ng et al., 2001). The $J$ global clusters are then assigned $J$ different labels, and a $J$-class hyperbolic SVM classifier is trained as follows. For $J(> 2)$ labels, one can use the common approach of training a $J$-class classifier by using $J$ independently trained binary classifiers. In particular, each of the $J$ binary classifiers corresponds to one of the $J$ training labels, and is trained independently on the same training set with the task of separating one chosen class from the remaining classes in the training data set. Thereafter, for each of the binary classifiers, the resulting prediction scores are computed for every data point and are transformed into probabilities using the Platt scaling technique (Platt et al., 1999). The predicted label for a given data point is obtained by using a maximum a posteriori criteria involving probabilities of the $J$ classes.

## I   Experimental Setup and Additional Results

### I.1   Data Sets

**Synthetic Data Sets.** We consider $R = 0.95$ and curvature constant $k = 1$ for all synthetic data sets, and choose $N = \mu \cdot 100,000$, for $\mu \in \{0.2, 0.4, 0.6, 0.8\}$. Each setting is tested via 10 independent trials. We use a 90%/10% random split for each data set to obtain training and test points.

**Biological data sets.** We consider the following data sets:

- **Olsson's scRNA-seq data set** (Olsson et al., 2016), containing single-cell (sc) RNA-seq expression data. It comprises 319 points from 8 classes (cell types). We use the Poincaré embeddings provided by (Chien et al., 2021) in their public code repository[3]. The curvature of the embedding is $k = 1$.

- **UC-Stromal data set**[4] from (Smillie et al., 2019), comprising cells from 68 colon mucosa biopsies from 18 ulcerative colitis patients and 12 healthy individuals. The cells were sequenced using 10X Chromium (either v1 or v2) and filtered to remove low-quality cells. We used a total of $26,678$ stromal and glia cells of dimension $1,307$, corresponding to the number of (highly) variable genes. We obtain the embeddings using mixed-curvature variational auto-encoders (m-VAE) (Skopek et al., 2020) with 1 MLP layer and 300 hidden dimensions. The curvature for the embeddings is set to $k = 0.0071$. The labels $0 - 3$ used in our results in Section 6 correspond to labels Myofibroblasts, Pericytes, Inflammatory Fibroblasts, and Glia, respectively.

- **Lung-Human-ASK440 data set**[5] from (Vieira Braga et al., 2019), comprising human lung cells from asthma patients and healthy controls. A total of $3,314$ cells from the parenchymal lung tissue were sequenced using Drop-seq protocol. The total dimension of the data points, corresponding to the number of genes, is $3,377$. Similar to UC-Stromal, we use m-VAE with 1 MLP layer and 300 hidden dimensions to obtain the Poincaré embeddings. The curvature constant for the embeddings is set to $k = 0.0015$. Furthermore, the labels $0 - 4$ correspond to labels Type-2, Type-1, T cell, NK cell, and Mast cell, respectively.

---

[3]https://github.com/thupchnsky/PoincareLinearClassification/tree/main/embedding.
[4]https://singlecell.broadinstitute.org/single_cell/study/SCP551/scphere. Registration is required to access the content.
[5]https://www.ncbi.nlm.nih.gov/geo/query/acc.cgi?acc=GSE130148. Accessed with GEO: GSE130148.

For each biological data set, we consider a 85%/15% random split for the training and test points, and keep it fixed for all trials. Our implementation for the m-VAE is provided at `https://github.com/thupchnsky/sc_mvae/tree/main`.

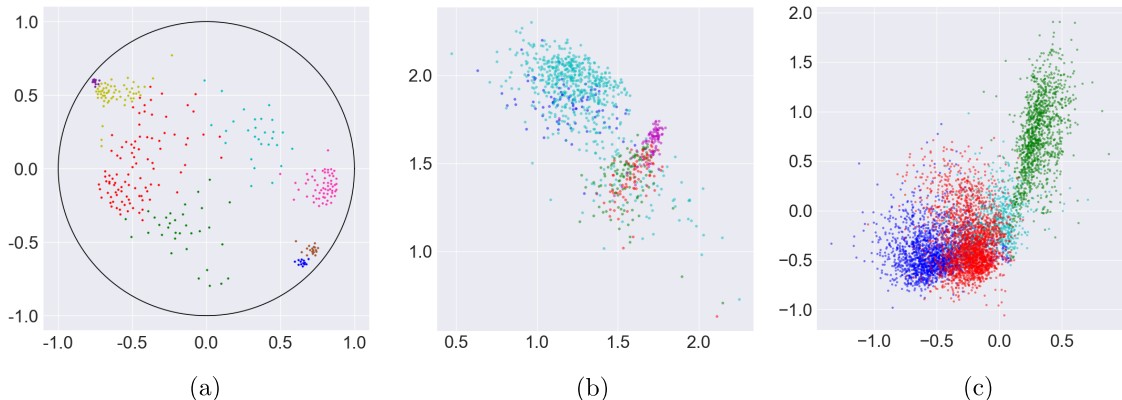

Figure 8: Visualization of the Poincaré embeddings for the three considered biological data sets. (a) Olsson's single-cell RNA expression. (b) Lung-Human-ASK440. (c) UC-Stromal. In (b) and (c), the boundary circle for the Poincaré disc has been omitted for better scaling for visualization purposes.

### I.2 Simulation Details

**Label Switching.** As the FL server applies our proposed graph partitioning based approach to aggregate the local convex hulls after recovering them, it does not need to know the ground truth labels or the exact local labels at each client (i.e., the $B_h$ labels suffice to recover the individual quantized convex hulls without knowing the exact identity of the clients from which they came). Hence, we did not simulate label switching in our code at any client because it does not affect the grouping procedure in our algorithm. After grouping the local convex hulls, the FL server assigns to each group a different label of its own choice. In our simulations, the ground truth training label associated with the majority of the local convex hulls in a given group is assigned as the label of the data points in that group before training the SVM classifiers on the recovered convex hulls. This is done so as to enable the computation of the accuracy of the trained classifiers at the FL server on the test data set in each setting.

**Graph-Based Hull Aggregation.** For binary classification, we use the kernighan_lin_bisection() module from the NetworkX library (Hagberg et al., 2008) in Python to group the convex hulls. For multi-label classification, we use the SpectralClustering() module from Scikit-learn (Pedregosa et al., 2011). Unlike kernighan_lin_bisection() which produces balanced partitions of nodes for the case of binary classification, SpectralClustering() based graph partitioning is not guaranteed to provide balanced partitioning for multi-label scenarios. To improve the partioning performance, we observe that the local hulls that belong to the same client should ideally be in separate groups. After $B_h$ decoding, the server can identify which of the local hulls come from the same client, albeit without knowing the identity of the specific client. Hence, in our simulations, we use a heuristic by which we assign very small weights to edges between each pair of local convex hulls from the same client, forcing SpectralClustering() to assign them to separate groups. This simple heuristic works very well in practice, as witnessed by our simulation results.

**Synthetic Data Sets.** For both Euclidean and Poincaré SVMs, we set the regularization hyperparameter in (7) to $\lambda = 20,000$, which essentially forces the solver to solve the hard margin SVM problem (6). For federated baselines, we consider $L = 10$, and partition the training data uniformly across clients.

**Biological Data Sets.** For both Euclidean and Poincaré SVMs, we consider a regularization term $\lambda = 0.1$. For federated baselines, we set $L = 3$, and partition the training data uniformly across clients. The default value for the quantization parameter (i.e., distance margin) is $\epsilon = 0.01$. For obtaining the reference points, we find that using the approach proposed in (Chien et al., 2021) (described in Section 3) may be unstable in some settings. Therefore, instead of selecting just the minimum distance pair, we select three pairs of points with lowest pairwise distances, and choose the one pair which produces the best training accuracy.

Table 2: Mean accuracy (%) and 95% confidence interval of the baselines for different data sets. The results are based on 10 independent trials for each setting.

| Data set | Labels | CP (%) | CH-CP (%) | CE (%) | CH-CE (%) |
|---|---|---|---|---|---|
| Olsson | 0-7 | $79.17 \pm 0.00$ | $83.33 \pm 0.00$ | $68.75 \pm 0.00$ | $77.08 \pm 0.00$ |
| Lung-Human | [0, 2, 3, 4] | $73.33 \pm 0.00$ | $74.07 \pm 0.00$ | $67.41 \pm 0.00$ | $68.89 \pm 0.00$ |

### I.3 Additional Results

**Federated Baselines Outperforming Centralized Ones.** For the rows in Table 1 where federated baselines offer better accuracy than their centralized counterparts, we carry out further analysis to examine the reasons behind the findings. For this, we consider additional centralized baselines where before training the SVM classifier (both for the Euclidean and Poincaré methods), we find the minimal convex hull for each class using Alg. 1. The SVM classifiers are then trained on the minimal convex hulls instead of the entire classes. The results are presented in Table 2, where we denote the baselines with minimal convex hulls by CH (Convex Hull). We can observe that training on minimal convex hulls indeed provides better performance than training on original points sets in this case.

Table 3: Mean accuracy (%) and 95% confidence interval of the baselines for different data sets. The results are based on 10 independent trials for each setting. The FL method with best mean accuracy is written in boldcase letters.

| Data set | Labels | CP (%) | CE (%) | FLP (%) | FLE (%) |
|---|---|---|---|---|---|
| UC-Stromal | 0-3 | $73.20 \pm 0.00$ | $73.54 \pm 0.00$ | $\mathbf{68.61 \pm 3.36}$ | $65.98 \pm 3.63$ |
| | [0, 1, 2] | $74.65 \pm 0.00$ | $73.94 \pm 0.00$ | $65.92 \pm 6.11$ | $\mathbf{67.14 \pm 5.67}$ |
| | [0, 1, 3] | $88.47 \pm 0.00$ | $87.39 \pm 0.00$ | $\mathbf{89.71 \pm 1.26}$ | $88.59 \pm 1.04$ |
| | [0, 2, 3] | $80.64 \pm 0.00$ | $82.29 \pm 0.00$ | $\mathbf{79.85 \pm 2.24}$ | $78.14 \pm 3.41$ |
| | [1, 2, 3] | $80.33 \pm 0.00$ | $80.33 \pm 0.00$ | $\mathbf{83.45 \pm 0.66}$ | $82.42 \pm 1.61$ |

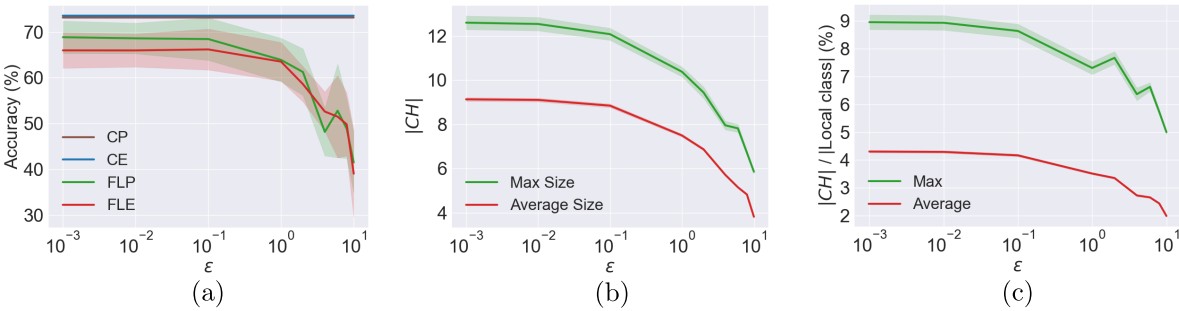

(a)      (b)      (c)

Figure 9: Impact of the quantization parameter $\epsilon$ on accuracy and convex hull complexity for UC-Stromal for the multi-label classification setting with labels $0-3$) and $L = 5$. Here, $CH$ denotes the quantized convex hull for an arbitrary class at an arbitrary client. The average as well as the max are calculated over all clients across all local quantized convex hulls. The shaded areas represent the 95% confidence interval from 10 independent trials.

**Results for $L = 5$.** Due to requiring the consent of patients for their data to be used for biological analyses, it is usually hard to compile large biological data sets. This is also the reason why the real-world data sets considered in this paper are (relatively) small. Furthermore, the number of participating clients with private biological data sets is expected to be small. Hence, in all our experiments from Section 6, we considered $L = 3$. Here, we provide additional results for $L = 5$ and the UC-Stromal data set, with $\epsilon = 0.01$. The accuracy results for various labels are presented in Table 3, while the results illustrating the impact of $\epsilon$ on the accuracy and the convex hull complexities for the setting with $[0-3]$ are shown in Figure 9. We observe

similar trends as for $L = 3$, except that the relative convex hull size increases notably, while the max still remains under 10%.

