# OpenReview forum: "Federated Classification in Hyperbolic Spaces via Secure Aggregation of Convex Hulls"
_TMLR — Accepted by TMLR_

### Review · Reviewer_Ce8E · 2023-08-27

**Summary Of Contributions:**

The paper proposes a one-shot federated SVM classification algorithm for hyperbolic spaces for analyzing hierarchical and tree-like data which cannot be embedded into Euclidean spaces of finite dimension with small distortion. Specifically, a distributed version of centralized convex SVM classifiers for Poincaré discs is proposed. Furthermore, the theoretical analysis of the expected complexity of random convex hulls of data sets in hyperbolic spaces is given. Experiments on synthetic data sets and biological data sets verify the effectiveness of the proposed algorithm.

**Audience:**

Yes

**Broader Impact Concerns:**

The study of this type of algorithm is a very interesting direction for the federated learning community

**Claims And Evidence:**

Yes

**Requested Changes:**

I am not familiar with this literature, and hence please weight other experts' opinion higher. The following are more high-level questions:

1. How exactly is privacy defined in the paper? What counts as a privacy leakage? Usually, we consider any form of access to raw data (not limited to accessing the data itself) to be a privacy leakage.

2. In Figure 5, the third row, it is interesting that when $\epsilon$ is relatively small, as $\epsilon$ increases, the accuracy rate does not change significantly. Can the author give some discussion?


3. Def 3 introduces $\epsilon>0$ as a parameter which is a little bit confusing to me since it does not appear in the definitions.

**Strengths And Weaknesses:**

Strength

1. A communication-efficient federated classification algorithm for analyzing hierarchical and tree-like data with theoretical analysis is proposed.

2. The numerical studies show the effectiveness of the proposed algorithm by comparing it with the state of the art (including its centralized version).

3. The paper is well-written and the expression is clear. It's easy to follow the main idea of the paper.

Weakness

1. It would be better to compare the communication cost and the computational cost with the state of the art in the experiments.

---

> ### Author Response · Authors · 2023-10-11
> **Author response**
>
> We thank the reviewer for their time and effort in reviewing the paper. We believe that the insightful comments will significantly strengthen the exposition of the results in the manuscript.
>
> ---
>
> 1. `Comparison with state-of-the-art methods with experiments`.
>
> We thank the reviewer for the very interesting questions raised. We would first like to point out that when it comes to federated classification in hyperbolic spaces, our work is the first of its kind, and there is no other method to directly compare it with, particularly due to the completely unresolved issue of label switching in other works. At the same time, there is a substantial body of work in FL, centered around securely transmitting the local models, but in that case, only information pertaining to local gradients/models is transmitted. We note that this is feasible since global labels for each local data point are assumed to be aligned across clients. In our case, it is not possible to rely on gradients, hence we use convex hulls of local classes instead. Nevertheless, in what follows, we discuss the computation and communication complexity of a gradient based approach for performing Euclidean SVM in the federated setting, assuming that there is no label switching issue.
>
> In line with the notations described in our paper, the global loss function for the \textbf{federated Euclidean problem} may be stated as follows:
>
> $\min_{\boldsymbol{w},b} \frac{1}{L} \sum_{i=1}^L F_i(\boldsymbol{w},b)$,
>
> where $F_i(\boldsymbol{w},b)$ denotes the local loss at client $i\in[L],$ which for Euclidean SVMs is defined as
>
> $F_i(\boldsymbol{w},b)=\frac{1}{2}||\boldsymbol{w}||^2+\frac{\lambda}{M_i}\sum_{j=1}^{M_i}\max(0,1-y_j(<\boldsymbol{w},\boldsymbol{x}_{i,j}>+b))$.
>
> Assume that there are $R$ training rounds, and that each client carries out $1$ epoch of local training in each communication round. For a given round $t\in[R]$, let $\boldsymbol{w}^{(G,t)}$ and $b^{(G,t)}$ denote the global weight and bias shared by the server with the clients. Assume for simplicity that client $i$ updates the global parameters using gradient descent,
>
> $\boldsymbol{w}^{(i,t)}=\boldsymbol{w}^{(G,t)}-\mu^{t} \frac{\partial F_i(\boldsymbol{w},b)}{\partial \boldsymbol{w}},$
>
> $b^{(i,t)}=b^{(G,t)}-\mu^{t} \frac{\partial F_i(\boldsymbol{w},b)}{\partial b},$
>
> where, $\mu^{t}$ denotes the learning rate at round $t$, $\frac{\partial F_i(\boldsymbol{w},b)}{\partial b}$ and $\frac{\partial F_i(\boldsymbol{w},b)}{\partial \boldsymbol{w}}$ denote the partial derivatives, i.e.,
>
> $\frac{\partial F_i(\boldsymbol{w},b)}{\partial \boldsymbol{w}}=\boldsymbol{w}-\frac{\lambda}{M_i}\sum_{j=1}^{M_i}\mathbb{1}[y_j (<\boldsymbol{w},\boldsymbol{x}_{i,j}>+b)<1] y_j \boldsymbol{x}_{i,j},$
>
> $\frac{\partial F_i(\boldsymbol{w},b)}{\partial b}=-\frac{\lambda}{M_i}\sum_{j=1}^{M_i}\mathbb{1}[y_j (<\boldsymbol{w},\boldsymbol{x}_{i,j}>+b)<1] y_j.$
>
> Here, $\mathbb{1}[\cdot]$ stands for the indicator function. After receiving the local models from the clients, the server can average the parameters to update the global model
>
> $\boldsymbol{w}^{(G,t+1)}=\frac{1}{\sum_{i=1}^L M_i}\sum_{i=1}^L M_i \boldsymbol{w}^{(i,t)}$
>
> $b^{(G,t+1)}=\frac{1}{\sum_{i=1}^L M_i}\sum_{i=1}^L M_i b^{(i,t)}.$
>
> Clearly, the computation cost for client $i$ is $O(M_i R)$. Similarly, the communication cost for client $i$ is $O(R)$. At server side, the overall computation complexity is $O(L R)$.
> In contrast, our approach does not use multiple rounds of communications (since the clients transmit their minimal convex hulls) and the clients do not perform any updates or computation beyond finding the convex hulls, quantizing the centroids and encoding the labels. For these steps, our computational and communication complexity results were reported in Section 4.5. We repeat them here for convenience:
>
> Communication complexity: $O(K_{i}L\max\{h\log L, \log B\})$ for client $i$, where $K_{i}$ (usually small compared with $M_i$) is the number of extremal points in the quantized convex hull of data points at client $i$, $L$ is the number of clients, $h$ is the maximum  number of clients sharing extremal points in the same quantized bin. $B$ is the number of quantization bins.
>
> Computation complexity: max{$O((K^{(i)})^2L\log (M_iL)),O(\max_{i\in[L]}[(K^{(i)}L)^{\frac{3}{2}}\log q+K^{(i)}L\log^2 q]),O(L^{h})$}, where $q=\max${$B,h\log L$}.

---

> > ### Author Response · Authors · 2023-10-11
> > **Author response**
> >
> > 2. `How exactly is privacy defined in the paper? What counts as a privacy leakage? Usually, we consider any form of access to raw data (not limited to accessing the data itself) to be a privacy leakage.`
> >
> > We thank the reviewer for the comment. We would like to clarify that in our setting, the notion of privacy pursued is described in Section 3.2, and it does not make use of classical notions of differential privacy and local differential privacy. Instead, it uses a new privacy constraint that is suitable for biological data. In a nutshell, biologists shun using differential privacy since biological data is highly noise and sparse and privacy noise tends to diminish the utility of the learners. This is why we adopted a more acceptable type of constraints. First, each client shares only the quantized versions of the minimum number of extreme points of the convex hulls of the local classes with the server. This sharing is done via secure aggregation so that the server has no information regarding the data provenance, i.e., the server cannot find which quantized convex hulls came from which client. Our privacy leakage notion is captured by the size of the local quantized convex hulls and is formally described in Definition 4. We have also characterized this privacy leakage by deriving the minimal number of points in a local quantized convex hull; the result for the same is described in Section 4.3, in particular Theorem 4.2, which shows that the convex hull complexity is small. The theoretical result matches our observations for real datasets, as demonstrated in Figure 7, where we can see that the quantized convex hulls are small in size.
> >
> > 3. `In Figure 5, the third row, it is interesting that when $\epsilon$ is relatively small, as $\epsilon$ increases, the accuracy rate does not change significantly. Can the author give some discussion?`
> >
> > We appreciate the reviewer's comment. When $\epsilon$ is small, our proposed framework introduces low distortion in the approximate global convex hulls for the different classes that are computed at the server side. This is because when $\epsilon$ is small, the quantization regions are small, and hence the local quantized convex hulls that are sent from each client closely approximate the true local convex hulls. Therefore, we do not incur significant losses in accuracy. We would also like to point out that the dependency of accuracy on $\epsilon$ is also impacted by the distribution of the data, which for Figure 5 is uniform. In Figure 7, we have presented results for demonstrating how $\epsilon$ can impact performance for real-world datasets. In particular, there is noticeable drop in accuracy when $\epsilon>0.1$.
> >
> > 4. `Def 3 introduces $\epsilon>0$ as a parameter which is a little bit confusing to me since it does not appear in the definitions.`
> >
> > We thank the reviewer for pointing this out. We would like to clarify that Definition 3 involves the definition of a Poincar\'e quantization scheme (please see Definition 2), which in turn has $\epsilon$ as a parameter. Hence, Definition 3 is also inherently dependent on $\epsilon$. To avoid any possible confusion, we have now changed the notation of the Poincar\'e quantization scheme from $PQ(\cdot)$ to $PQ_{\epsilon}(\cdot)$ everywhere, including Definition 3, to make this dependency explicit. The newly added text is marked in blue for quick reference.

---

### Review · Reviewer_a9cx · 2023-09-12

**Summary Of Contributions:**

The paper introduces a novel federated learning (FL) method tailored for hyperbolic spaces, specifically for the Poincaré disc model, to handle tree-like data sets more efficiently than traditional Euclidean methods. This approach is crucial for hierarchical data like single-cell RNAseq data, which often comes with privacy constraints. The new method combines convex support vector machine (SVM) classifiers, number-theoretic techniques for label alignment, and quantization methods for efficient data communication. Tests show the hyperbolic FL method outperforms its Euclidean counterpart in classification accuracy, especially in biomedical data scenarios.

**Audience:**

Yes

**Claims And Evidence:**

Yes

**Requested Changes:**

See above

**Strengths And Weaknesses:**

**Strengths**
1) The paper presents a compelling motivation for working with RNAseq data, which is hierarchical in nature and where strong privacy constraints exist.
2) Several techniques, such as Poincare Quantization, are discussed that could be of independent interest.


**Questions/Weakenesses**

1) Paper considers *linear* SVM classifiers on Poincare disks. Did the authors try different non-linear kernels ? If it is non-trivial there should be discussion about this.
2) Why did authors not consider Riemannian Federated methods (Riemannian Optimization in Federated setup) as potential baselines ? For example see works like  [1,2]
3) Supposed improvement (11%) is modest when compared to Euclidean FL, especially given the increased complexity in the overall procedure. However, this minor remark holds true for all solutions operating in a Riemannian setting.


**Refs**:

[1] Li, J. and Ma, S., 2022. Federated Learning on Riemannian Manifolds.

[2] Wu, X., Hu, Z., & Huang, H. (2023). Decentralized riemannian algorithm for nonconvex minimax problems.

---

> ### Author Response · Authors · 2023-10-11
> **Author response**
>
> We thank the reviewer for their time and effort in reviewing the paper. We believe that the insightful comments will significantly strengthen the exposition of the results in the manuscript.
>
> ---
>
> 1. `Extension to non-linear kernels`.
>
> We appreciate the insightful comment from the reviewer. In the centralized learning context, adapting to non-linear kernels is straightforward. Since we have access to all data points, the logarithmic mapping from the hyperbolic space to the tangent space reduces the problem to a conventional Euclidean SVM classification involving the entire training set. As a result, we can subsequently use off-the-shelf kernel SVM techniques to accommodate non-linear kernels. However, in the federated setting, the problem becomes significantly more complex. Here, the server is restricted from directly accessing individual data points. Instead, the server only receives aggregated local convex hulls from clients. While kernel SVM remains feasible at the server, relying solely on convex hulls might compromise the performance of the kernel SVM algorithms. Therefore, effectively integrating non-linear kernels within our current framework that relies on convex hull approaches may be challenging. On the other hand, if one could dispose of the use of convex hulls, then potentially non-linear kernels could be potentially be used, but then we would not know how to resolve the underlying data privacy issues. We believe that the combination of convex hulls and SVM represents the best trade-off in terms of the privacy+utility performance.
>
> We would like to emphasize that we recognize that exploring non-linear kernel methods is a compelling idea. However, addressing this problem in-depth lies beyond the current scope of our study.
>
> 3. `Riemannian Federated methods as potential baselines.`
>
> We would like to thank the reviewer for pointing us to these two recent papers on Riemannian manifold learning in the FL setting, and we have now added these references into our revision (please see blue text in Section 2). However, we would like to clarify the distinction between our work and these two references, and show that it is actually non-trivial to consider these two methods as potential baselines.
>
> First, while hyperbolic spaces are Riemannian manifolds, our paper aims to formulate the hyperbolic SVM classification problem as a \textbf{convex optimization} problem that is solvable through Euclidean optimization methods. In contrast, [1, 2] emphasize the extension of traditional Euclidean optimization algorithms, such as SVRG, to Riemannian manifolds within the FL framework. Thus, our focus is more on a new problem formulation, while [1, 2] primarily address the optimization procedure.
>
> Second, given that our problem is reformulated as a convex optimization task in Euclidean spaces, integrating the approaches from [1, 2] would not improve the performance of our algorithm. For these methods to be fully effective, one needs to define SVM classification as an optimization problem directly in hyperbolic spaces, as done in [3], and subsequently considered in [1, 2]. Notably, the methodologies in [1, 2] require understanding the exponential mapping and parallel transport specific to the target manifold. In [1, 2] the authors only consider the Stiefel manifold, and it is non-trivial to combine the methods with the problem defined in [3], where the manifold is defined by $w * w > 0$ ($*$ stands for the Lorentzian inner product).
>
> Third, we would like to emphasize that due to label switching, prior federated schemes cannot be directly compared to ours. Particularly, they make the (practically unrealistic) assumption that the global labels for each local data point are known in advance. Our framework is the first to resolve this issue by leveraging combinatorial $B_h$ sequences.
>
> We agree that incorporating Riemannian optimization in Federated Learning setting to solve hyperbolic space SVM problem is definitely an exciting and promising future direction. Nonetheless, given the breadth and depth of such an endeavor, it remains beyond the scope of the current paper.
>
>
>
>
>
>
>
>
>
> ---
>
> [1] Li, J. and Ma, S., (2022). Federated learning on Riemannian manifolds.
>
> [2] Wu, X., Hu, Z., Huang, H. (2023). Decentralized Riemannian algorithm for nonconvex minimax problems.
>
> [3] Cho, Hyunghoon, et al. (2019). Large-margin classification in hyperbolic space.

---

### Review · Reviewer_Hxr3 · 2023-09-30

**Summary Of Contributions:**

For context for my review, I am knowledgeable on hyperbolic spaces and their use in machine learning, I have some prior experience with differential privacy, but I do not know too much about federated learning.

This paper looks at federated learning in hyperbolic space. Concretely, suppose we want to train a classifier on data, but we cannot do this in a centralized manner. Instead, our data is distributed among $L$ different *clients*. Hence we want to collect data from the clients and do the computations but do so in a "private" manner.

The paper then presents a protocol whereby each client encodes their data, then sends this data to the centralized server. The server then learns a classifier on the data received by the server. The contributions of the paper are

0) The framework for federated learning
1) An algorithm for finding the extreme points of convex hulls in Hyperbolic space
2) A way of tiling Hyperbolic space
3) The theory result on the number on the expected number of extreme points of a convex hull in Hyperbolic space
4) Experiments showing reasonable results.

**Audience:**

No

**Claims And Evidence:**

No

**Requested Changes:**

My primary request is could you please clarify the setting and the goal? Specifically, I would really appreciate a definition of privacy to help motivate some of the steps in the pipeline.

**Strengths And Weaknesses:**

**Strengths**
---

The procedure here consists of many different steps. Each step by itself is interesting and the authors have interesting results for each step.

**Weaknesses**
---

1) The primary weakness for me is that the paper is not clear. I do not mean that the sentence to sentence writing is not clear, but the message of the paper is not clear.

    a) The first point that is really not clear to me is - what is the actual problem? The problem seems to be that we have distributed information and want a way to learn a classifier. However, I do not understand why we can't collect the data as is and then train a standard classifier. The paper mentions issues related to privacy and computational complexity, but I have two issues with this. 1) privacy is never formally defined. Hence it is very hard to see how privacy is preserved. 2) The authors quantize to reduce complexity, but I think the additional extra computations in relation to the having to recompute the extreme points, as well as the intricate label decoding might defeat this benefit.

    b) Following form this, I think each step in the process could also be better motivated. I understand the authors have tried to provide diagrams for the sequence of events taking place, but it is still quite difficult to follow. It wasn't until I got to the balanced grouping part did I understand why it was ok to encode the labels.

    c) The final bit that is confusing to me is why are we training an SVM. As far as I understand it, we have a fixed data. We do one round of communications and then we classify the data. Hence the grouping could be the classification. Why do we then need to train an SVM on top of that? Is there a test set that is recommunicated in the same way?

**Questions**
---

I have a few questions

1) Do we think of Z as a collection of tuples? Specifically, I wanted clarification on the non-intersecting assumptions? Can we have two clients with the $z_{I,j}$ but if they have different labels? From the rest of the paper I think so.

2) How did you make the choice of embedding algorithms? Have you tried any of the hierarchical embedding techniques such as Sarkar 2013, De Sa, Gu, Re, Sala 2018, Sonthalia, Gilbert 2020, and Lin, Coifman, Mishne, Talmon 2023?

3) I don't think the quantization scheme is the first one. Please see Yu and De Sa, 2019

4) I am confused as to why we need to recompute the convex hull. I assume this has to do with this fact that the reference point has to be outside this true convex hull. But it is unclear to me whether recomputing the convex hull fixes this.

---

> ### Author Response · Authors · 2023-10-10
> **Author response**
>
> We thank the reviewer for their time and effort in reviewing the paper. We believe that the insightful comments will significantly strengthen the exposition of the results in the manuscript.
>
> 1. `The actual problem`.
>
> Thank you for the question. The short answer is that we are concerned with federated classification in the hyperbolic space where the server learns a classification model in hyperbolic space from the local client data, while ensuring client data privacy in a manner suitable for biological data.
>
> Since the reviewer is an expert in learning in hyperbolic spaces and differential privacy, but according to their own comment, not so familiar with federated learning, we would first like to clarify the federated learning aspects of our problem. This will result in a rather long answer, but hopefully also clarify our contributions.
>
> In the federated learning setting, we have to devise a distributed classification method for hyperbolic spaces, solve the label switching problem associated with distributed learning, and devise privacy-preserving and efficient client-server communication protocols that do not add noise to the data (such as differential privacy). The latter is crucial for biological data analysis, because biological data is highly noisy and computational biologists do not fully embrace noise-adding mechanisms which tend to significantly reduce the utility of the models applied to them.
>
> To reiterate, the main motivation behind this work comes from genetic/multi-omic data analysis where data often comes in hierarchical form and is traditionally stored in a distributed manner. Examples of this are hospitals or data depositories that cannot freely exchange private patient data. Besides the privacy considerations (which will be discussed in detail in the next part of our response), another reason why we do not directly collect data and let the server train the model is the communication and computational complexity. Transmitting all the data from clients to the server is insurmountable in terms of communication costs (as anecdotal evidence, experts from Mayo clinic have shared their stories of using FedEx carriers to share data on hard disc drives with collaborating hospitals since the volume of the data prohibited direct electronic sharing). Therefore, we need communication-efficient and privacy-preserving schemes for hierarchical data.
>
> In addition to reducing the communication cost and preserving privacy, there is the rather substantial and (we believe) mostly unsolved problem of label switching, which arises in federated settings when different clients label their data in different manner. Imagine that one client labels data from one class as $+1$ while another client labels data from the same class as $-1$. Then, data points of this class received from these two clients will be considered as coming from different classes. To address such label switching issues, we used tools from number theory (e.g., $B_h$ sequences) to encode the labels, which is one of the pivotal technical contributions of our study. This allows the server to recover the quantized convex hulls of the local classes at each client, without learning anything about the data provenance (since the cient's data is securely aggregated before it reaches the server). More precisely, the server cannot associate the identities of the clients to the recovered convex hulls. Furthermore, the server receives only a very small number (this number is analytically quantified in the paper) of quantized extreme points of the convex hull of the data sets at the client side, and the quantization noise may be roughly viewed as a noise-privacy mechanism, although without the classical differential privacy guarantees. We believe that this approach is the one most suitable for biological applications. Finally, to complete the federated learning approach we also had to design a novel graph based technique for grouping the recovered convex hulls, which is then followed by learning the global SVM classifier.

---

> > ### Author Response · Authors · 2023-10-11
> > **Author response**
> >
> > 2. `What is the formal definition of privacy`.
> >
> > We appreciate the reviewer's inquiry. Our privacy considerations do not align with conventional paradigms of differential privacy (DP) or information theoretic privacy. Rather than adhering to a singular mathematical definition of privacy, we have adopted and combined multiple privacy criteria, as explained in Section 3.2. We believe that this notion of privacy is most suitable for computational biology applications, where differentially private noise is considered undesirable as it significantly reduces the utility of learning methods.
> >
> > Therefore, our notion of privacy asks for what is in between the requirements of DP and information theoretic privacy. Specifically, we require that 1) The clients reveal a small as possible number of data points that are quantized (both for the purpose of reducing communication complexity and further obfuscating information about the data). 2) The server collects data without knowing its provenance or the individual clients that contribute the data through the use of secure aggregation of the quantized minimal convex hulls of the client data.
> >
> > Accordingly, we quantify privacy through the definition of privacy leakage (Def. 3 in page 7 of the paper), which aims to settle the first requirement. Our client-server communication scheme (SCMA) hides the data provenance, which satisfies the third requirement.
> >
> > 3. `The motivation of each step in the process`.
> >
> > As detailed in our response to Question 1, each step of our process is driven by specific motivations, some of which we will reiterate for clarity. Our primary focus is on federated classification within hyperbolic space, in the context of biological data processing. When contrasted with conventional centralized classification in Euclidean space, our approach grapples with three distinctive challenges.
> >
> > First, privacy. Our specific privacy definition can be found in our response to Question 2. To ensure our method adheres to this privacy standard, we advocate for the identification of minimum convex hulls at clients, client-side data quantization, and the utilization of the SCMA scheme for secure data aggregation at the server level. Comprehensive details are provided on page 8 of our paper.
> >
> > Second, label switching. Each client can have different local labels for data points that comes from the same ground truth class. To resolve this issue, we use $B_h$ sequence encoding at the client side and decoding at the server side. Subsequently, balanced grouping on graphs is used to cluster the local convex hulls. Further information is available on page 7 of our paper.
> >
> > Third, finding the reference point to perform the classification task at the server side is nontrivial. While the first two challenges are inherent to general FL problems, the third challenge arises due to the nature of the hyperbolic SVM algorithm we chose to adopt, which necessitates a reference point for the logarithmic map. Given that we can not straightforwardly average the local reference points to deduce a global one, our approach involves transmitting the local convex hulls to the server. This allows for the computation of an approximate global reference point directly at the server. A more detailed exposition is available on page 6 of our paper.
> >
> > In summary, each step of our approach is tailored to address one or more of these three challenges. We hope that this explains our approach, but please let us know if further explanations are needed.
> >
> > 4. `Why train an SVM at the server after grouping?`.
> >
> > As highlighted in our response to Question 1, the primary objective is for the server to derive a classification model from distributed training data, rather than simply classify the relayed data. Consequently, it's imperative for the server to train an SVM classifier and create a global model. With this model in place, upon receiving new test data, the server can either directly classify the test data or, if needed, share the trained model with the clients to facilitate their own classification tasks. A real-world application of this can be in the context of data sharing with the National Institutes of Health (NIH), where models are trained using data from various participating hospitals in a federated learning approach. Subsequently, these models are disseminated to the public, aligning with one of the primary NIH mission goals to safely share data models that can be adapted and retested (i.e., reproduced).

---

> > > ### Author Response · Authors · 2023-10-11
> > > **Author Response**
> > >
> > > 5. `Do we think of Z as a collection of tuples?`
> > >
> > > We thank the reviewer for this question. Yes, $Z$ is a collection of tuples, including the data and the labels. By "non-intersecting" we mean that no two clients share the same data with the same label. We note that local datasets being non-intersecting is a simplifying assumption that is commonly found in standard federated learning works. It is also an assumption in line with the type of application we pursue - most patients only go to one hospital and have their data stored in one repository. Nevertheless, our protocols are applicable even when clients have overlapping local datasets. Also, as the reviewer pointed out correctly, two clients with the same data can have different local labels, though their global labels can be the same.
> > >
> > > 6. `How did we make the choice of embedding algorithms?`
> > >
> > > While the embedding process is not the primary focus of this work (given our assumption that clients have already embedded their data into the Poincar\'e disc), we did not prioritize fine-tuning this procedure. As we describe in our experiments in Section 5, for synthetic datasets, we generated datasets directly within the Poincare disk. For real datasets, we utilized Mixed-curvature Variational Autoencoders (m-vae) for embedding into the Poincar\'e disk, mostly because of its well-maintained implementation on GitHub. That said, we recognize the importance of referencing the other relevant embedding literature as pointed out by the reviewer. In our revision, we have included appropriate citations for the ones we missed previously (please see the blue text in Data Model and Poincar\'e Embeddings in Section 3.2).
> > >
> > > 7. `The quantization scheme is the first one`.
> > >
> > > We appreciate the reviewer's comment. The paper \[Yu, Tao, and Christopher M. De Sa. "Numerically accurate hyperbolic embeddings using tiling-based models." Advances in Neural Information Processing Systems 32 (2019).\] indeed introduces what they term "tiling-based models." This approach does reduce to quantization of the hyperbolic space, but is distinct from our proposed quantization scheme. During our literature review, our search primarily revolved around the keyword "quantization," which led to our oversight regarding a proper citation of this work. We have added a reference to this paper in our revised manuscript (please see the blue text in our main contributions in Section 1), and have also removed the claim that ours is the first known quantization scheme.
> > >
> > > 8. `Why we need to recompute the convex hull.`
> > >
> > > There are two points at which we (re)compute the convex hull: 1) After the balanced grouping, the **server** will find the global convex hull to train a hyperbolic SVM classifier; 2) After finding the convex hull and quantization, the **client** will recompute the convex hull of the quantized extremal points.
> > >
> > > For case 1): As outlined in our response to Question 4, the server's objective after balanced grouping is to train a hyperbolic SVM classifier in preparation for future incoming test data. For this training process, the server need to find the global convex hulls for groups of local convex hulls, then select the most suitable reference point(s).
> > >
> > > For case 2): We wish to underscore the rationale behind our three-phase process (computing the convex hull, quantizing the extremal points, then recomputing the convex hull of these quantized points). This process is designed to reduce the convex hull complexity, which reduces both communication costs and potential privacy leakage. Although this process might seem computationally demanding, the double computation of the convex hulls is a deliberate trade-off. We are prioritizing reduced communication and enhanced privacy, both of which are paramount in FL scenarios, over computational simplicity.

---

> > > > ### Comment · Reviewer_Hxr3 · 2023-10-16
> > > > **Thank you for the response**
> > > >
> > > > I sincerely thank the authors for their detailed response. I know better understand the problem and setting.

---

### Author Response · Authors · 2023-10-11
**Author response**

We thank all the reviewers for their careful reading and insightful comments. We have revised our paper accordingly and the changes are marked in blue.

---

### Author Response · Authors · 2024-01-01
**Thanks and uploading the revised manuscript**

Dear Action Editor,

We want to express our sincere gratitude for your time and effort, and handling the reviews of our paper. We also acknowledge the useful feedback received by the reviewers. We have addressed your request regarding the minor revisions, and have uploaded the revised manuscript. All relevant changes in our revised manuscript are marked in blue. We have also shared our codebase on GitHub and have provided the link to the public repository in our abstract.

1. `The response in https://openreview.net/forum?id=umggDfMHha&noteId=sWngboaJXb included a detailed discussion of the problem that paper seeks to solve. It would be useful to have at least some part of this in the paper, e.g., in Section 1 or Section 3.2.`

    We have integrated the response into our revised manuscript. In particular, we now describe the main motivation for our problem of federated classification in hyperbolic spaces at the beginning of Section 4 (previously Section 3.2). For convenience, we have included the write-up below for quick reference:

    *As described in Section 1, our approach is motivated by genomic/multiomic data analysis, since such data often exhibits a hierarchical structure and is traditionally stored in a distributed manner. Genomic data repositories are subject to stringent patient privacy constraints and due to the sheer volume of the data, they also face significant communication and computational complexity overheads.*

    *Therefore, in the federated learning setting corresponding to such scenarios, one has to devise a distributed classification method for hyperbolic spaces that is privacy-preserving and allows for efficient client-server communication protocols. Moreover, due to certain limitations of (local) differential privacy (DP), particularly with regards to biological data, new privacy constraints are required. DP is achieved through injection of privacy noise, which is problematic because biological data is already highly noisy and adding noise significantly reduces the utility of the inference pipelines. Furthermore, the proposed method needs to address potential ``label switching'' problems, which arise due to inconsistent class labeling at the clients, preventing global reconciliation of the clients' data sets at the server side.*

    Additionally, on page 8 in the revised manuscript, we have provided a more detailed discussion of the Label Switching problem, a critical issue left mostly unaddressed in prior works on federated learning. For convenience, we are including the write-up below:

    *Label Switching. Another major challenge faced by the clients is ``label switching.'' When performing classification in a distributed manner, unlike for the centralized case, one has to resolve the problem of local client labels being potentially switched compared to each other and/or the ground truth. Clearly, the mapping from the local labels at the client level to the global ground truth labels is unknown both to the client and the server.*

    *The label switching problem is illustrated in Figure 2, where clients use mismatched choices for the binary labels of their local data points. To enable federated classification, the label switching problem has to be addressed in a private and communication efficient manner, which is challenging. Our proposed solution is shown in Figure 2, and it is based on $B_h$ sequences. In a nutshell, $B_h$ sequences are sequences of positive integers with the defining property that any sum of $\leq h$ possibly repeated terms of the sequence uniquely identifies them (see Section 5.4 for a rigorous treatment). Each client is assigned two different integers from the $B_h$ sequence to be used as labels of their classes, so that neither the server nor the other clients have access to this information. Since secure aggregation is typically performed on sums of labels, and the sum uniquely identifies the constituent labels that are associated with the client classes, all labels can be matched up with the two ground truth classes. As an illustrative example, the integers $1, 3, 7, 12, 20, 30, 44, \ldots$ form a $B_2$ sequence, since no two (not necessarily distinct) integers in the sequence produce the same sum. Now, if one client is assigned labels $1$ (red) and $3$ (green), while another is assigned $7$ (blue) and $12$ (pink), then a securely aggregated label $1+7=8$ (purple) for points in the intersection of their convex hulls uniquely indicates that these points belonged to both the cluster labeled $1$ at one client, and the cluster labeled $7$ at another client (see Figure 2, right panel).*

    Furthermore, we have now provided a more comprehensive discussion of our privacy model. For details, please see our next response.

---

> ### Author Response · Authors · 2024-01-01
> **Thanks and uploading the revised manuscript (continued)**
>
> 2. `Include a clear statement of the paper's notion of privacy and how the proposals satisfy this. E.g., in Section 1 or Section 3.2, based on the response in https://openreview.net/forum?id=umggDfMHha&noteId=P4Pdjew51N.` `Include discussion of why (Local) DP is not favoured in the specific problem setting, e.g., at end of Section 2 or in "Privacy leakage" paragraph. Include a comment on whether, in different applications where federated hyperbolic classification may be natural, there are still reasons to favour the paper's notion of privacy compared to (Local) DP. [The title refers to FL for generic hyperbolic classification, so it would be good to clarify whether the paper's notion of privacy is suggested to be used in all such applications.]`
>
>      In our proposed method, each client shares only the quantized versions of the minimum number of extreme points of the convex hulls of the local classes with the server. Hence, our privacy leakage notion is captured by the size of the local quantized convex hulls and is formally described in Definition 4 in our manuscript. In the following, we are including Definition 4 for convenience.
>
>     *Definition 4. ($\epsilon$-Convex hull complexity) Given $\epsilon>0$, a $\epsilon$-Poincar\'e quantization scheme ${PQ_{\epsilon}}(\cdot)$, and a set $\mathcal{D}$ of $N$ points in the Poincar\'e disc, let $Q=$\{${PQ_{\epsilon}}(\boldsymbol{x}):\, \boldsymbol{x}\in CH(\mathcal{D})$\} be the point set obtained by quantizing the extreme points in the convex hull of $\mathcal{D}$. The $\epsilon$-convex hull complexity is the number of extreme points in the convex hull of $Q$, i.e., the size of $\hat{CH}(\mathcal{D})$.*
>
>     The notion of privacy used in our approach is new, and captured in terms of the convex hull complexity. In a nutshell, the larger the convex hull, the higher the privacy leakage and vice versa. We have characterized the privacy leakage more formally by deriving an expression for the expected minimal convex hull; the result is available in Section 5.3. In particular, Theorem 5.2 establishes that the expected minimal convex hull complexity is small. We are including Theorem 5.2 for quick reference.
>
>     *Theorem 5.2. Let $\boldsymbol{x}_1,\ldots,\boldsymbol{x}_N$ be $N$ points uniformly distributed over a Poincar\'e disc of radius $R<s$, where $N$ is sufficiently large. Let $\hat{\boldsymbol{x}}_1,\ldots,\hat{\boldsymbol{x}}_N$ be quantized values of $\boldsymbol{x}_1,\ldots,\boldsymbol{x}_N$, obtained using the quantization rule described in (13), with parameter $\epsilon=O(N^{-c}),$ for some constant $c>0$. Then, the expected number of extreme points of the minimal convex hull of $\hat{\boldsymbol{x}}_1,\ldots,\hat{\boldsymbol{x}}_N$ is at most $O(N^{c})$ when $c< \frac{1}{2}$, at most $O(N^{1-c})$ when $\frac{1}{2}\le c\le \frac{2}{3}$, and at most $O(N^{\frac{1}{3}})$ when $\frac{2}{3}< c$.*
>
>     Furthermore, we have now included more detailed explanations on the privacy notions considered in our paper. Additionally, we have also explained our rationale regarding differential privacy approaches. Our privacy notions are applicable in other distributed data processing scenarios, as also explained in our revised manucript. Details can be found in the highlighted text after Definition 4. For convenience, we are including them below.
>
>     *We would point out that our notion of privacy leakage is not based on (local) differential privacy (DP) or information theoretic privacy. Instead, it may be viewed as a new form of ``combinatorial privacy'' enhanced by controlled quantization-type obfuscation. This approach to privacy is most suitable for computational biology applications, where DP noise is considered undesirable as it significantly reduces the utility of learning methods.*
>
>     *In addition to compromising accuracy, DP has other practical drawbacks in many general applications (Kifer \& Machanavajjhala, 2011; Bagdasaryan et al., 2019; Zhang et al.,  2022; Kan, 2023). Particularly, Bagdasaryan et al. (2019) established that DP models are more detrimental for underrepresented classes. In fact, fairness disparity is significantly pronounced in DP models, as the accuracy drop for minority classes is more significant than for other classes. Additionally, in many practical scenarios, meeting requirements for DP, which rely on particular assumptions about the data probability distribution, might prove challenging (Kifer \& Machanavajjhala, 2011; Zhang et al., 2022; Kan, 2023). For instance, when database records exhibit strong correlations, ensuring robust privacy protection becomes difficult. All of the above issues are inherent to biological (genomic) data. Specifically, genomic data is imbalanced across classes. Furthermore, as it derives from human patients, it is highly correlated due to controlled and synchronized activities of genes within cells of members of the same species.*

---

> > ### Author Response · Authors · 2024-01-01
> > **Thanks and uploading the revised manuscript (continued)**
> >
> > 2. `Include a clear statement of the paper's notion of privacy and how the proposals satisfy this. E.g., in Section 1 or Section 3.2, based on the response in https://openreview.net/forum?id=umggDfMHha&noteId=P4Pdjew51N.` `Include discussion of why (Local) DP is not favoured in the specific problem setting, e.g., at end of Section 2 or in "Privacy leakage" paragraph. Include a comment on whether, in different applications where federated hyperbolic classification may be natural, there are still reasons to favour the paper's notion of privacy compared to (Local) DP. [The title refers to FL for generic hyperbolic classification, so it would be good to clarify whether the paper's notion of privacy is suggested to be used in all such applications.]` (continued)
> >
> >      *To overcome the aforementioned concerns, we require that 1) The clients reveal as small as possible number of data points that are quantized (both for the purpose of reducing communication complexity and further obfuscating information about the data). 2) The server collects data without knowing its provenance or the individual clients that contribute the data through the use of secure aggregation of the quantized minimal convex hulls of the client data. To address the first requirement, we introduce a new notion of privacy leakage in Definition 4. If the number of quantized extreme points is small and the quantization regions are large, the $\epsilon$-convex hull complexity is small and consequentially the privacy leakage is ``small'' (i.e, the number of points whose information is leaked is small). For bounds on the $\epsilon$-convex hull complexity in hyperbolic spaces (and, consequently, the privacy leakage), the reader is referred to Section 5.3. Quantization introduces noise that further obfuscates information about client data. To address the second requirement, we adapt standard secure aggregation protocols to prevent the server from inferring which convex hull points belong to which client, thus hiding the data provenance.*
> >
> > 3. `Include discussion of the potential issues (or lack thereof) with transmitting extreme points of the quantised convex hulls, e.g., whether there is a risk they might correspond to transmitting patients with easily identifiable records. If there is no issue here, that is ideal; if there is a potential issue, it would be good to mention and note as a challenge in hyperbolic FL for future work to study.`
> >
> >     While we minimize sharing information from a client to the server via use of quantized convex hulls, we agree with the AE that there could be scenarios where extreme points of the quantized convex hulls could contain important identifiable information of sensitive ``outlier patients". Even with differential privacy protocols, outliers are hard to handle -- either in terms of ensuring the same privacy level or the same degree of utility. Furthermore, these points require special processing and analysis and have to be communicated to the server in some shape or form. We have now included a discussion of this issue in the Conclusion section (please see our response to Bullet 6 below for quick reference to the Conclusion section). However, as we have pointed out in our response to Bullet 2, using other privacy mechanisms such as (local) differencial privacy (DP) can degrade the model accuracy due to added noise. Furthermore, due to the disparate effect of DP on minorities as compared to majority classes, extremal points (which are outliers in the data sets) are already in danger of having much lower accuracy when protected via DP. Hence, we believe that it is better to share the smallest number of extremal points with the server rather than use DP-like mechanisms.
> >
> > 4. `Include discussion of whether there might be other ways of achieving privacy beyond the paper's proposal.`
> >
> >     As already described in our response within Bullet 2, other privacy mechanisms such as (local) differential privacy (DP) could be used instead of quantized minimal convex hulls. However, due to multiple limitations of DP in the context of genomic data classifications, we believe that our proposed privacy approach is more applicable for the problem at hand.
> >
> > 5. `(Optional) Consider whether Section 3.2 should belong under "Preliminaries": it is 4.5 pages, and gives quite a lot of detail about the individual steps in the procedure. Indeed, the sub-section is about as long as Section 4. It might be nice to consider re-arranging material across Section 3.2 and Section 4.`
> >
> >     We have converted Section 3.2 into a separate section. In particular, Section 3 now exclusively covers the preliminaries regarding classification in hyperbolic spaces. On the other hand, Section 4 now includes the problem formulation, a list of questions as well as their solutions.

---

> > > ### Author Response · Authors · 2024-01-01
> > > **Thanks and uploading the revised manuscript (continued)**
> > >
> > > 6. `(Optional) consider having an explicit concluding section to discuss future work.`
> > >
> > >     We have added a Conclusion section. For convenience, we are including it below:
> > >
> > >     *We introduced a novel approach to federated learning in hyperbolic spaces, thereby addressing challenges in processing hierarchical and tree-like data in distributed and private settings. Specifically, we developed an end-to-end framework for federated learning of SVM classifiers in the Poincaré disc. The core of the approach it to leverage securely aggregated convex hulls for information transfer from the clients to the server. Furthermore, the complexity of convex hulls is analyzed to assess data leakage, and a simple quantization method is proposed for efficient data communication. We also considered detrimental label switching issues and addressed it with a new method based on number-theoretic and coding-theoretic ideas. Additionally, we introduced a novel approach for aggregating client convex hulls using balanced graph partitioning. Experimental results on multiple single-cell RNA-seq data show improved classification accuracy compared to Euclidean counterparts, establishing the utility of privacy-preserving learning in hyperbolic spaces.*
> > >
> > >     *Our work also introduces many potentially important new research questions. While information sharing via quantized convex hulls is efficient and secure, extreme points of the quantized convex hulls can contain important identifiable information, and addressing this limitation is an important problem for future work. Additionally, characterizing the impact of quantization on privacy leakage and generalizing the privacy-preserving and communication efficient federated learning protocols to other ML tasks are other questions of interest.*

---

### Decision · Action_Editor_aE3W · 2023-12-05

**Recommendation:** Accept with minor revision

**Comment:**

The paper proposes a new approach for federated classification in hyperbolic space, which necessitates new mechanisms for communicating information from server to client, accounting for label switching, and aggregating information at the server.

The initial reviews found the paper to address an interesting problem, with a number of novel technical contributions. There were also some concerns raised, primarily on the following points:

(1) comparison to Riemannian FL methods

(2) clarity on the precise problem setting and constraints, to help motivate the need for the various components of the proposed solution

(3) the precise notion of privacy used in the paper, how and why it deviates from DP

For (1), the response clarified the differences between the paper's setting and those of recent Riemannian FL works (which focus on the optimisation procedure), and the non-triviality of applying such techniques to federated classification in hyperbolic spaces.

For (2), the response gave a more detailed explanation of the problem setting.

For (3), the response gave more clarity on the paper's notion of privacy. However, there is scope for the manuscript to more clearly discuss some points:

- the paper does not use (Local) DP. In the response, the authors explain the rationale for this in the context of biological applications. Such discussion would be useful to include in the text.

- the proposed scheme requires transmitting certain extreme points of quantised convex hulls from client to server. The privacy leakage is taken to be the total number of transmitted points. Typically, communicating even a few individual records is not favoured in privacy settings. Perhaps this is a fundamental challenge in hyperbolic FL (owing to the requirement of knowledge of convex hulls) that could be important to study in future work; but at the least, there could be more discussion of this point. e.g., there is not much discussion on the what such extreme points might plausibly correspond to in the setting of interest (anomalous patient records, which might be sensitive?).

- the paper's notion of privacy appears entwined with the communication protocol. e.g., it is not clear whether there might be other strategies that are also private under the paper's definition.


Overall the paper makes a number of technical contributions that could be of interest to the community. Some edits to clarify the notion of privacy could help better contextualise the work, qualify potential limitations of the approach, and suggest directions for future work.



**Suggested changes**

We suggest the following changes prior to publication. Most of these involve incorporating text from the authors' response.

- the response in https://openreview.net/forum?id=umggDfMHha&noteId=sWngboaJXb included a detailed discussion of the problem that paper seeks to solve. It would be useful to have at least some part of this in the paper, e.g., in Section 1 or Section 3.2.

- include a clear statement of the paper's notion of privacy and how the proposals satisfy this. e.g., in Section 1 or Section 3.2, based on the response in https://openreview.net/forum?id=umggDfMHha&noteId=P4Pdjew51N.

- include discussion of why (Local) DP is not favoured in the specific problem setting, e.g., at end of Section 2 or in "Privacy leakage" paragraph.

- include a comment on whether, in different applications where federated hyperbolic classification may be natural, there are still reasons to favour the paper's notion of privacy compared to (Local) DP. [The title refers to FL for generic hyperbolic classification, so it would be good to clarify whether the paper's notion of privacy is suggested to be used in _all_ such applications.]

- include discussion of the potential issues (or lack thereof) with transmitting extreme points of the quantised convex hulls, e.g., whether there is a risk they might correspond to transmitting patients with easily identifiable records. If there is no issue here, that is ideal; if there is a potential issue, it would be good to mention and note as a challenge in hyperbolic FL for future work to study.

- include discussion of whether there might be other ways of achieving privacy beyond the paper's proposal.

- (*Optional*) consider whether Section 3.2 should belong under "Preliminaries": it is 4.5 pages, and gives quite a lot of detail about the individual steps in the procedure. Indeed, the sub-section is about as long as Section 4. It might be nice to consider re-arranging material across Section 3.2 and Section 4.

- (*Optional*) consider having an explicit concluding section to discuss future work.

**Audience:**

Classification in hyperbolic spaces is of growing interest. Doing so in settings with de-centralised training is of import. The paper makes a number of non-trivial technical contributions to the problem that are likely of interest to the community.

**Claims And Evidence:**

The paper includes a number of technical components that are explained at both a high-level, and with more fine-grained details. While most of the claims are clear, there were some lingering concerns regarding the paper's precise notion of privacy remaining elusive. These would be good to address in a revision (see Comments).